# Rho-associated kinase (ROCK) function is essential for cell cycle progression, senescence and tumorigenesis

**Sandra Kümper[1]\*, Faraz K Mardakheh[1], Afshan McCarthy[1], Maggie Yeo[1], Gordon W Stamp[2], Angela Paul[1], Jonathan Worboys[3], Amine Sadok[1], Claus Jørgensen[3], Sabrina Guichard[1], Christopher J Marshall[1]†**

[1]Division of Cancer Biology, Institute of Cancer Research, London, United Kingdom; [2]Experimental Pathology Laboratory, Cancer Research UK London Research Institute, London, United Kingdom; [3]Cancer Research UK Manchester Institute, Manchester, United Kingdom

**Abstract** Rho-associated kinases 1 and 2 (ROCK1/2) are Rho-GTPase effectors that control key aspects of the actin cytoskeleton, but their role in proliferation and cancer initiation or progression is not known. Here, we provide evidence that ROCK1 and ROCK2 act redundantly to maintain actomyosin contractility and cell proliferation and that their loss leads to cell-cycle arrest and cellular senescence. This phenotype arises from down-regulation of the essential cell-cycle proteins CyclinA, CKS1 and CDK1. Accordingly, while the loss of either *Rock1* or *Rock2* had no negative impact on tumorigenesis in mouse models of non-small cell lung cancer and melanoma, loss of both blocked tumor formation, as no tumors arise in which both *Rock1* and *Rock2* have been genetically deleted. Our results reveal an indispensable role for ROCK, yet redundant role for isoforms 1 and 2, in cell cycle progression and tumorigenesis, possibly through the maintenance of cellular contractility.

\*For correspondence: sandra.kuemper@icr.ac.uk

†Deceased

**Competing interests:** The authors declare that no competing interests exist.

## Introduction

The Rho-associated coiled-coil-containing protein serine/threonine kinases ROCK1 and ROCK2 are downstream effectors of the Rho subfamily of small GTPases. They are activated by interaction with Rho GTPases and act through a number of pathways to regulate the actin cytoskeleton and thus cell migration, cell-cell adhesion and cancer cell invasion (*Itoh et al., 1999*; *Olson and Sahai, 2009*; *Thumkeo et al., 2013*). Substrates for ROCKs include the myosin-binding subunit of myosin phosphatase (MYPT1) and the myosin regulatory light chain (MLC), which regulate actomyosin contractile forces (*Kawano et al., 1999*; *Kureishi et al., 1997*). Actomyosin contractility is generated when myosin regulatory light chain (MLC) is phosphorylated, allowing myosin II interaction with actin filaments to generate mechanical force (*Aksoy et al., 1983*; *Kureishi et al., 1997*). ROCKs may either directly phosphorylate MLC (*Amano et al., 1996*) or indirectly regulate MLC phosphorylation by phosphorylating the myosin-binding subunit of myosin phosphatase (MYPT1) and inactivating it (*Kimura et al., 1996*). Myosin phosphatase inactivation results from phosphorylation of two inhibitory sites, Thr696 and Thr850, on MYPT1 (*Feng et al., 1999*; *Kawano et al., 1999*; *Muranyi et al., 2005*). Furthermore, ROCKs phosphorylate and activate LIM kinases 1 and 2 (LIMK1, 2), which inhibit the actin-depolymerizing protein Cofilin (*Yang et al., 1998*). ROCK1 and 2 exhibit 65% overall identity and 87% within the kinase domain, and some studies suggest differential roles for the isoforms. Using RNA interference, ROCK1 was reported to be important for stress fiber formation in fibroblasts, whereas ROCK2 controls cortical contractility and phagocytosis (*Yoneda et al., 2005*;

**eLife digest** Animal cells contain a structure called the cytoskeleton, which helps give the cells their shape. This structure can rapidly disassemble and reassemble, which enables cells to change their shape, move and divide into two. Many proteins are involved in controlling these processes. In particular, two proteins called ROCK1 and ROCK2 are known to be important for helping cancer cells move. However, investigations into the exact roles of these proteins have so far produced contradictory results.

Kümper et al. have now developed a more refined method of studying what ROCK1 and ROCK2 do in cells. This involves genetically engineering mice in a way that makes it possible to control whether ROCK1 and ROCK2 are produced in specific cell types and tissues. Studying cells that had been taken from these mice revealed that cells that lacked both proteins could not contract. Moreover, these cells became bigger and flattened out. This change in appearance went hand in hand with the cells becoming unable to divide and form new cells. However, cells that lacked just one type of ROCK protein were still able to divide and proliferate.

As tumors form as a result of cells dividing and proliferating uncontrollably, Kümper et al. then studied how the ROCK proteins affect tumor development in mice that are susceptible to lung or skin cancer. Although cancerous cells were found that contained just one type of ROCK protein, no tumor cells were found that lacked both ROCK1 and ROCK2, further confirming that having one ROCK protein is essential for tumor formation.

Overall, it appears that within the systems studied ROCK1 and ROCK2 perform the same roles, and that ROCK proteins are indispensable for cell proliferation and hence tumor development. The next challenge will be to identify the tumor types that are highly dependent on processes driven by the ROCK proteins. Further work could then investigate whether drugs that inhibit the activity of the ROCK proteins block the growth and spread of these tumors.

*2007*). Distinct roles for ROCK1 and 2 have also been described in the regulation of keratinocyte differentiation (*Lock and Hotchin, 2009*) and cell detachment (*Shi et al., 2013*).

Mice in which *Rock*1 has been genetically deleted are born, but have defects in eyelid as well as ventral body wall closure (*Shimizu et al., 2005*). Ninety percent of *Rock2* null mice die in utero due to defects in the placental labyrinth layer. This indicates that ROCK1 cannot compensate for a loss of ROCK2. However, the few *Rock2* null mice, that are born, display defects similar to those described in *Rock1* null mice (*Thumkeo et al., 2003*). This indicates some level of functional redundancy (*Thumkeo et al., 2005*). In addition to their role in cell migration, ROCKs have been reported to modulate apoptosis (*Coleman et al., 2001*; *Sebbagh et al., 2005*) and cell proliferation (*Croft and Olson, 2006*; *Samuel et al., 2011*; *Zhang et al., 2009*). The precise role of ROCKs in cell proliferation is not clear: some reports suggest ROCK function is required for G1/S progression (*Croft and Olson, 2006*; *Zhang et al., 2009*), but others suggest ROCK is only required for anchorage-independent growth of transformed cells (*Sahai et al., 1999*; *Vigil et al., 2012*). One in vivo study reported that over-activation of ROCK, by expressing the kinase domain of ROCK2 in mouse skin, led to hyperproliferation and epidermal thickening (*Samuel et al., 2011*). In order to investigate the roles of ROCK1 and 2 in tumorigenesis, we have generated conditional *Rock1* and *2* knockout mice and studied these in vivo, using genetically engineered mouse models of non-small cell lung cancer (NSCLC) and *Braf*^V600E mutant models of melanoma, as well as in vitro, using cells isolated from these mice.

## Results

### Depletion of both ROCK1 and 2 leads to multi-stage cell cycle arrest through defects in actomyosin contractility

It is well understood that ROCKs control key aspects of the actin cytoskeleton such as actomyosin contractility, but their role in cell proliferation is not known. Because *Rock* null mice die early due to developmental defects, we generated *Rock1* and *2* conditional alleles (*Rock1*^f and *Rock2*^f) by

inserting LoxP sites flanking exon 6 in the *Rock1* locus and exons 5 and 6 in the *Rock2* locus (*Figure 1—figure supplement 1A*). These exons are located within the kinase domain and their deletion results, through frameshifts, in the absence of ROCK proteins. We first generated cells lacking ROCK1 (*Rock1$^{\Delta/\Delta}$*), ROCK2 (*Rock2$^{\Delta/\Delta}$*) or ROCK1 and 2 (*Rock1$^{\Delta/\Delta}$;Rock2$^{\Delta/\Delta}$*) by isolating mouse embryo fibroblasts (MEFs) from embryos with the following genotypes: *Rock1$^{f/f}$;Rock2$^{wt/wt}$, Rock1$^{wt/wt}$;Rock2$^{f/f}$* or *Rock1$^{f/f}$;Rock2$^{f/f}$* and infecting them with Adenovirus-expressing Cre recombinase (Ad-Cre) or GFP (Ad-GFP). Depletion of ROCK1 and ROCK2 (*Rock1$^{\Delta/\Delta}$;Rock2$^{\Delta/\Delta}$*) resulted in a strong impairment of cell proliferation in vitro, which was not observed in cells lacking either ROCK1 (*Rock1$^{\Delta/\Delta}$*) or ROCK2 (*Rock2$^{\Delta/\Delta}$*) (*Figure 1A*), demonstrating that ROCK function is required for cell proliferation and that ROCK1 and ROCK2 act redundantly. Infection of cells with adenovirus is not 100% efficient, and when the growth of *Rock1$^{\Delta/\Delta}$;Rock2$^{\Delta/\Delta}$* cells was monitored over a longer period of time, these cells eventually recovered their ability to proliferate (*Figure 1—figure supplement 1B*), but western blot analysis revealed that these cells express ROCK1 and 2 in equivalent levels to wild type cells (data not shown) and thus likely originated from uninfected cells.

As genetic depletion abrogates ROCK function long term, we investigated whether long-term treatment of cells with ROCK inhibitors caused proliferation defects. Cells treated for 48 hr with the ROCK inhibitor H1152 (*Sasaki et al., 2002*) had reduced proliferation (*Figure 1B*). Similar results were observed with other ROCK inhibitors, such as GSK269962A, AT13148, GSK429286A and chroman1 (data not shown). However, the much-used ROCK inhibitor Y-27632 (*Narumiya et al., 2000*) had a much weaker effect on cell proliferation than H1152 (*Figure 1—figure supplement 1C*). This is consistent with previous studies, where we have shown that Y-27632 is a less effective ROCK inhibitor than H1152 (*Sadok et al., 2015*). To determine whether defects in proliferation were due to ROCK-mediated effects on actomyosin contractility, cells were treated with blebbistatin, an inhibitor of myosin II ATPase, for 48 hr. Treated cells displayed a similar proliferation defect to that observed in *Rock1$^{\Delta/\Delta}$;Rock2$^{\Delta/\Delta}$* cells (*Figure 1B*), arguing that the requirement for ROCK in cell proliferation is mediated through its role in maintaining actomyosin contractility.

To test whether *Rock1$^{\Delta/\Delta}$, Rock2$^{\Delta/\Delta}$* or*Rock1$^{\Delta/\Delta}$;Rock2$^{\Delta/\Delta}$* MEFs can proliferate in vivo, MEFs isolated from *Rock1$^{f/f}$, Rock2$^{f/f}$* or *Rock1$^{f/f}$;Rock2$^{f/f}$*mice were immortalized with retrovirus expressing a dominant-negative form of Trp53 (Trp53 DD) (*Hahn et al., 2002*), and subsequently transformed with a retrovirus expressing mutant H-Ras (H-Ras$^{V12}$). The cells were then treated with Ad-Cre or control virus and, 5 days later, injected subcutaneously and tumor growth monitored. No differences in tumor growth were observed when transformed MEFs, derived from *Rock1$^{f/f}$*or *Rock2$^{f/f}$* cells, infected with Ad-Cre were compared to control virus (*Figure 1C,D*). In contrast, transformed *Rock1$^{\Delta/\Delta}$;Rock2$^{\Delta/\Delta}$* MEFs grew significantly slower in vivo compared to controls (*Figure 1E*). Western Blot analysis revealed the retention of ROCK1 and 2 protein in tumors derived from *Rock1$^{\Delta/\Delta}$;Rock2$^{\Delta/\Delta}$* (Trp53 DD; H-Ras$^{V12}$) MEFs but complete deletion of ROCK1 in *Rock1$^{\Delta/\Delta}$* and ROCK2 in *Rock2$^{\Delta/\Delta}$* tumor samples (*Figure 1F*, data not shown), in agreement with the observed results on proliferation in vitro. This result shows that injected *Rock1$^{\Delta/\Delta}$;Rock2$^{\Delta/\Delta}$* cells are unable to proliferate and form tumors but the un-infected cells can proliferate and form tumors thereby accounting for the delay in tumorigenesis and implying that ROCK function is essential for proliferation and hence tumor initiation and formation.

We here show that concomitant depletion of ROCK1 and ROCK2 leads to defects in cell proliferation in vitro as well as in vivo. Depletion of a single ROCK isoform however has no effect, suggesting they can act redundantly.

## ROCK1 and ROCK2 act redundantly to regulate actomyosin contractility and cell shape

In order to identify causes for the defects in cell proliferation seen in cells lacking ROCK1 and ROCK2 (*Rock1$^{\Delta/\Delta}$;Rock2$^{\Delta/\Delta}$*), we characterized their cell morphology. One of the most prominent responses to activated ROCK is the generation of actin stress fibers, which regulate cell shape (*Amano et al., 1997*; *Ridley, 1999*). A previous report using RNA interference suggested that ROCK1, rather than ROCK2, is important for stress fiber formation (*Yoneda et al., 2005*). After thorough analysis of ROCK depletion upon infection with Ad-Cre or Ad-GFP, a slight decrease in protein levels was observed after three days (*Figure 2—figure supplement 1A*). After four days a further reduction was seen and after five days both ROCK1 and 2 were depleted (*Figure 2B–C*). Three days after infection with Ad-Cre, *Rock1$^{\Delta/\Delta}$;Rock2$^{\Delta/\Delta}$* cells lacked stress fibers and displayed a 'tail

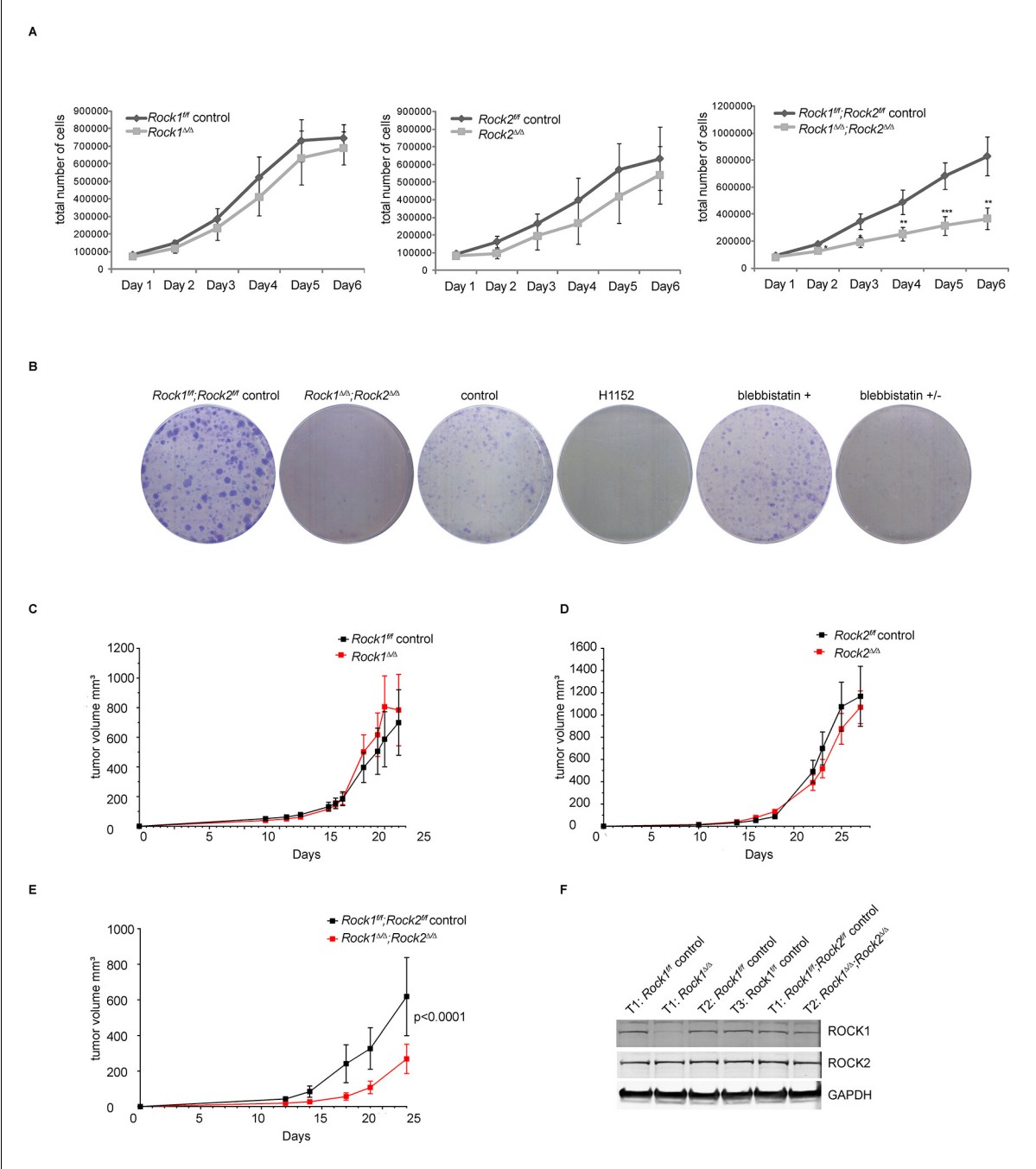

**Figure 1.** Depletion of ROCK1 and 2 leads to defects in cell proliferation in vitro and in vivo. (**A**) Proliferation curves of MEFs with different genotypes over 6 days. The cells were seeded 3 d after adenovirus infection. Graphs show total number of cells and SD from 5 independent experiments each carried out in triplicates. p-values were calculated using Student's t-test: ** p<0.005; *** p<0.001. (**B**) $Rock1^{f/f};Rock2^{f/f}$ control and $Rock1^{\Delta/\Delta};Rock2^{\Delta/\Delta}$ MEFs were cultured for 3 days and wild-type cells were treated with H1152, inactive blebbistatin (+) or active blebbistatin (+/-) for 48 hr. Cells from all conditions were then subjected to a colony formation assay and grown for a further 7 days. (**C–F**) $Rock1^{f/f};Rock2^{f/f}$ MEFs transformed with Trp53 DD and HRas V12 were treated with Ad Cre to generate Δ. Cells were injected subcutaneously into CD1 nude mice and growth analyzed. The graph shows average tumor volume in mm$^3$ and SEM for $Rock1^{\Delta/\Delta}$ and control (**C**), $Rock2^{\Delta/\Delta}$ and control (**D**), $Rock1^{\Delta/\Delta};Rock2^{\Delta/\Delta}$ and control (**E**). p values were calculated by ANOVA and are as indicated. (**F**) Tumors with stated genotypes were immunoblotted with indicated antibodies.

The following figure supplement is available for figure 1:

**Figure supplement 1.** Generation of conditional *Rock* alleles and cell proliferation analyses.

retraction' phenotype, a well-known characteristic of ROCK inhibition (*Somlyo et al., 2000*). Cells had long processes and defects in detaching their tails, in contrast to their controls (*Figure 2A* and *Videos 1,2*). A similar phenotype was observed in cells treated with the ROCK inhibitor H1152 or the myosin II inhibitor blebbistatin but not in cells lacking either ROCK1 (*Rock1*$^{\Delta/\Delta}$) or ROCK2 (*Rock2*$^{\Delta/\Delta}$) (*Figure 2A*). After five days, cells that lack both ROCK1 and ROCK2 had few apparent central stress fibers, unlike control cells, and surprisingly adopted a flat morphology (*Figure 2A* and *Videos 1,2*). This phenotype seemed to appear gradually after prolonged loss of ROCK which was seen three days after adenovirus infection. Interestingly, when random motility was analyzed between Days 3 and 4 after adenovirus infection, using ImageJ to track cells, migration speed and directionality were significantly increased in cells lacking ROCK1 and 2 (*Figure 2—figure supplement 1B*). This was previously described by Lomakin et *al.* (*Lomakin et al., 2015*).

Consistent with the levels of stress fibers, phosphorylation of MYPT and MLC were only affected in cells where both ROCKs had been removed (*Figure 2B–C*, quantification in *Figure 2—figure supplement 1C*), showing that ROCK1 and ROCK2 act redundantly in regulating MYPT and MLC phosphorylation. While MYPT Thr850 phosphorylation was markedly reduced in *Rock1*$^{\Delta/\Delta}$;*Rock2*$^{\Delta/\Delta}$ cells, MYPT Thr696 phosphorylation was not (*Figure 2B*, quantification in *Figure 2—figure supplement 1C*), suggesting that, while Thr850 is principally regulated by ROCKs, Thr696 is regulated by other kinases such as MRCK (*Wilkinson et al., 2005*). ROCK1 levels were slightly but not significantly increased in *Rock2*$^{\Delta/\Delta}$ cells and vice versa (*Figure 2—figure supplement 1C*).

*Rock1*$^{\Delta/\Delta}$;*Rock2*$^{\Delta/\Delta}$ MEFs spread when plated on a thick layer of collagen 5 days after adenovirus infection, unlike *Rock1*$^{\Delta/\Delta}$ or *Rock2*$^{\Delta/\Delta}$ cells which had a more spindle-like morphology similar to wild-type controls (*Figure 2D*). This indicates reduced cellular contractility upon loss of ROCK1 and 2. As a functional measure of actomyosin contractility, we used a gel contraction assay (*Hooper et al., 2010*), demonstrating that cells containing only ROCK1 or ROCK2 were able to contract a collagen gel, but cells lacking both ROCK1 and ROCK2 were not (*Figure 2E*), supporting the notion ROCK1 and ROCK2 act redundantly to maintain cellular contractility.

To further investigate ROCK signaling, we analyzed the temporal profile of changes in the phospho-proteome upon ROCK inhibition using mass spectrometry. Short-term inhibition of ROCK, using H1152, resulted in a significant decrease in phosphorylation of many phospho-sites, including several known targets of ROCK such as Cofilin-1 (Ser-3), Cofilin-2 (Ser-3), Destrin (Ser-3), and Ser-19/Thr-20 of MLC (*Supplementary file 1* and *Figure 2—figure supplement 1D*). However, the majority of these phosphorylation sites, with the exception of MLC Ser-19/Thr-20, were only transiently affected, as their phosphorylation was fully recovered in long-term ROCK inhibited cells (*Supplementary file 1* and *Figure 2—figure supplement 1E*) or *Rock1*$^{\Delta/\Delta}$;*Rock2*$^{\Delta/\Delta}$ cells (*Figure 2B–C*). These results indicate that alternative phosphorylation mechanisms exist for some ROCK substrates, but that no other kinase previously implicated in MLC phosphorylation can substitute for ROCK1/2 in the generation of actomyosin contractility.

In summary, we showed that short-term (three days after Ad-Cre infection) depletion of ROCKs induces a previously described 'tail retraction' phenotype but surprisingly long-term (from five days after Ad-Cre infection) depletion induces a further change in morphology resulting in a flattened morphology and decrease in central stress fibers.

## Loss of ROCK-dependent actomyosin contractility leads to cellular senescence

The flattened and enlarged morphology observed upon long-term depletion of ROCK1 and ROCK2 is typical of cellular senescence. Indeed, *Rock1*$^{\Delta/\Delta}$;*Rock2*$^{\Delta/\Delta}$ MEFs, or MEFs treated with H1152 or blebbistatin for prolonged periods exhibited a significant increase in staining for senescence-associated β-galactosidase (SA-βGal) (*Debacq-Chainiaux et al., 2009*) than control cells (*Figure 3A*). Senescence was also observed after prolonged treatment with other ROCK inhibitors such as GSK269962A, AT13148, GSK429286A and chroman1 (*Figure 3—figure supplement 1A*). To determine whether abrogation of ROCK function causes senescence in vivo, we used the recently described inhibitor AT13148 that has been shown to potently inhibit ROCK and has been previously used in vivo (*Sadok et al., 2015*; *Yap et al., 2012a*). 690cl2 mouse melanoma cells were injected intra-dermally into CD1 athymic mice, and the mice treated with AT13148 at 40 mg/kg. Senescence was detected by SA-βGal staining of tumors. We observed a decrease in tumor growth when mice were treated with AT13148 compared to controls (*Figure 3—figure supplement 1B*). Additionally,

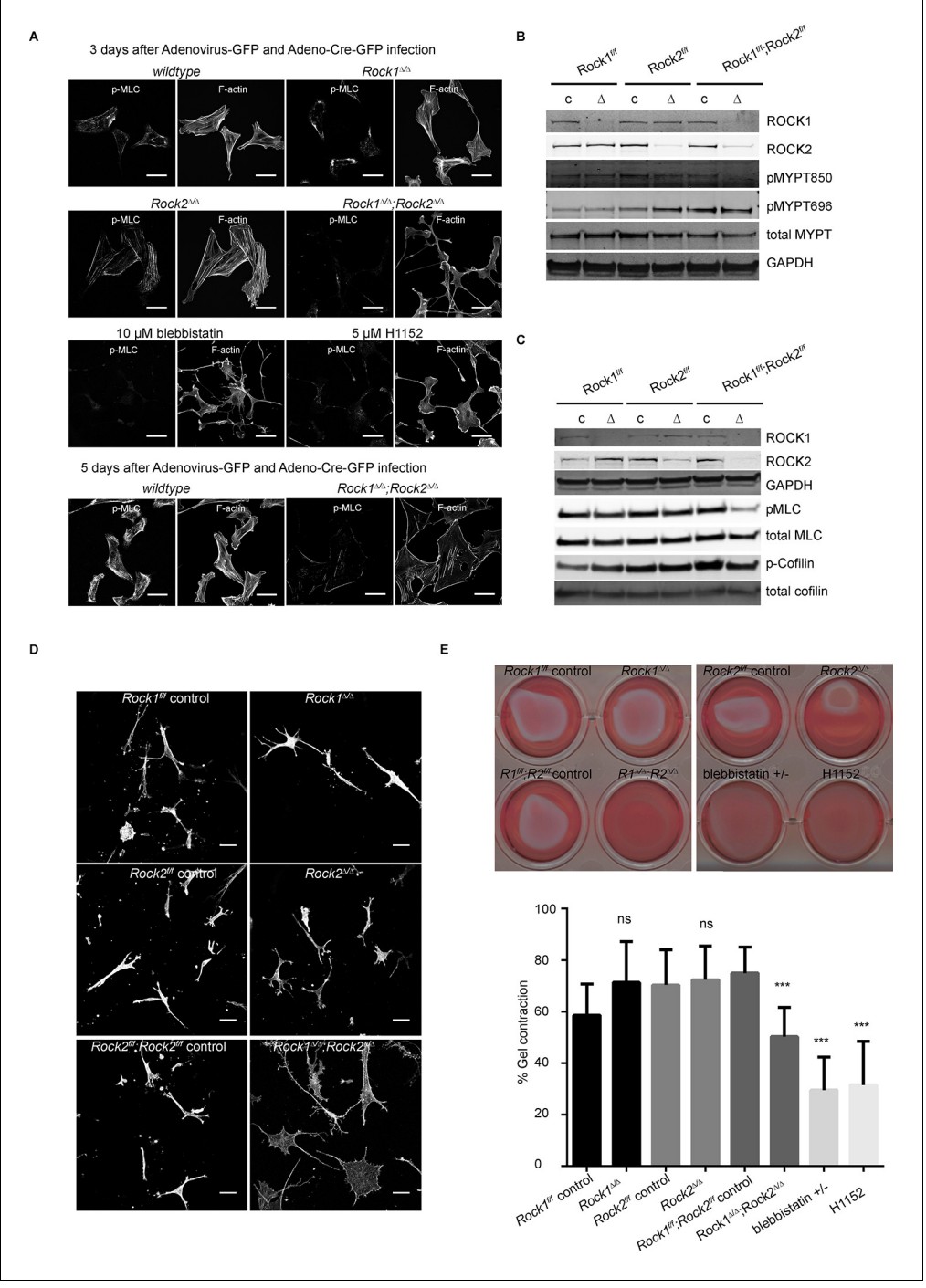

**Figure 2.** Only deletion of both ROCK1 and 2 leads to a loss of central stress fibers and a decrease in actomyosin contractility. (**A**) Images of wild-type, $Rock1^{\Delta/\Delta}$, $Rock2^{\Delta/\Delta}$ or $Rock1^{\Delta/\Delta};Rock2^{\Delta/\Delta}$ MEFs, 3 days after Ad-GFP and Ad-Cre-GFP infection, stained for pMLC and phalloidin. Wild-type and $Rock1^{\Delta/\Delta};Rock2^{\Delta/\Delta}$ cells are also shown 5 days after Ad-GFP and Ad-Cre-GFP infection. Scale bars are 50 μm. (**B, C**) Western blot analyses of ROCK targets in lysates of MEFs with indicated genotypes. Representative blots are shown, quantification of multiple biological replicates can be found in *Figure 2—figure supplement 1C*. (**D**) Images of MEFs with indicated genotypes plated on a thick layer of collagen and stained for phalloidin. Scale bars are 50 μm. (**E**) Images and quantification of collagen gel contraction assay using MEFs of indicated genotypes. Graph shows average data and SD from 5 independent experiments, each carried out in triplicate. p-values were calculated using Student's t-test: ***p<0.001.

*Figure 2 continued on next page*

*Figure 2 continued*

The following figure supplement is available for figure 2:

**Figure supplement 1.** Depletion of ROCK leads to an increase in migration speed and persistent down-regulation of Myosin Regulatory Light Chain (Mrlc) phosphorylation.

we observed an increase in the number of SA-βGal positive cells within tumors (*Figure 3—figure supplement 1C*).

Analysis of $Rock1^{\Delta/\Delta}$;$Rock2^{\Delta/\Delta}$ cells by microscopy revealed a significant increase in the number of cells with two nuclei (*Figure 3B*), indicating a failure of cytokinesis. During cytokinesis an actomyosin-based contractile ring is formed to drive cleavage furrow ingression, which subsequently results in two daughter cells. ROCK and the related Citron kinase have been shown to localize to the cleavage furrow, yet it is not entirely clear which kinase regulates MLC phosphorylation at the furrow to generate actomyosin contractility (*Kosako et al., 1999*; *2000*; *Madaule et al., 1998*). Analysis of time-lapse movies further confirmed a role for ROCK1 and 2 in cytokinesis as a greater percentage of $Rock1^{\Delta/\Delta}$;$Rock2^{\Delta/\Delta}$ cells failed to divide compare to wild-type cells (*Figure 3C* and *Videos 3,4*). In addition to an increased number of binucleate cells, analysis of time-lapse movies of $Rock1^{\Delta/\Delta}$; $Rock2^{\Delta/\Delta}$ cells revealed an overall lower number of dividing cells (*Figure 3D* and *Videos 3,4*). These data suggest that cells depleted of ROCK1 and 2 do not initiate division further, thus indicating a cell cycle block.

To analyze the cell cycle profile of $Rock1^{\Delta/\Delta}$;$Rock2^{\Delta/\Delta}$ cells, cells were stained with propidium iodide (PI) along with control cells for analysis by flow cytometry. Fewer double knockout than wild-type MEFs were in G1 or S phase, and a higher percentage of cells in G2/M phase of the cell cycle (*Figure 3E*). The increased percentage may be due to the fact that binucleate cells in G1 will have the same DNA content as cells in G2/M. To investigate a potential cell cycle block, we treated $Rock1^{\Delta/\Delta}$;$Rock2^{\Delta/\Delta}$ cells and their controls with nocodazole, commonly used to arrest cells with a G2/M phase DNA content (*Blajeski et al., 2002*). In controls, nocodazole led to an increased proportion of cells with G2/M DNA content as expected (*Figure 3—figure supplement 1D*). In $Rock1^{\Delta/\Delta}$; $Rock2^{\Delta/\Delta}$ cells, however, distribution of cells in the three cell cycle phases is very similar between control treated and nocodazole treated cells (*Figure 3—figure supplement 1D*). These data suggest that ROCK depletion causes a cell cycle block prior to G2/M. As there is an overall decrease in S phase cells, the block is likely to be in the G1 phase of the cell cycle.

## ROCK regulates cell cycle proteins CKS1 and CDK1

We have shown that ROCK depletion or inhibition induces cellular senescence; therefore, we next investigated whether they affected key cell cycle proteins involved in senescence. Two major pathways are activated by senescence signals. Activation of the transcription factor Trp53 induces expression of p21, a cyclin-dependent kinase inhibitor (CDKI) (*Brown et al., 1997*). The other pathway is induction of the CDKI p16, which prevents CDK4/6-mediated phosphorylation of

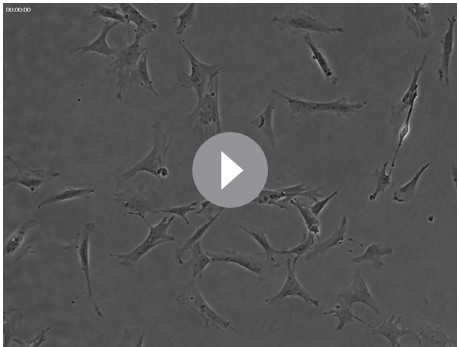

**Video 1.** $Rock1^{f/f}$;$Rock2^{f/f}$ control, 3 days after infection with Ad-GFP, 10X magnification.

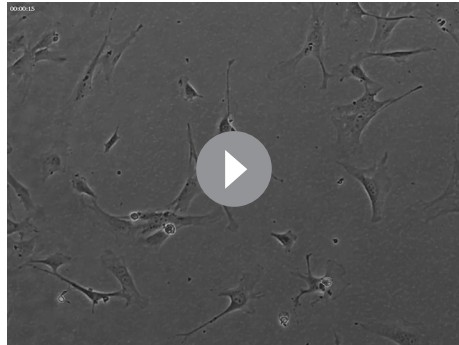

**Video 2.** $Rock1^{\Delta/\Delta}$;$Rock2^{\Delta/\Delta}$, 3 days after infection with Ad-Cre, 10X magnification.

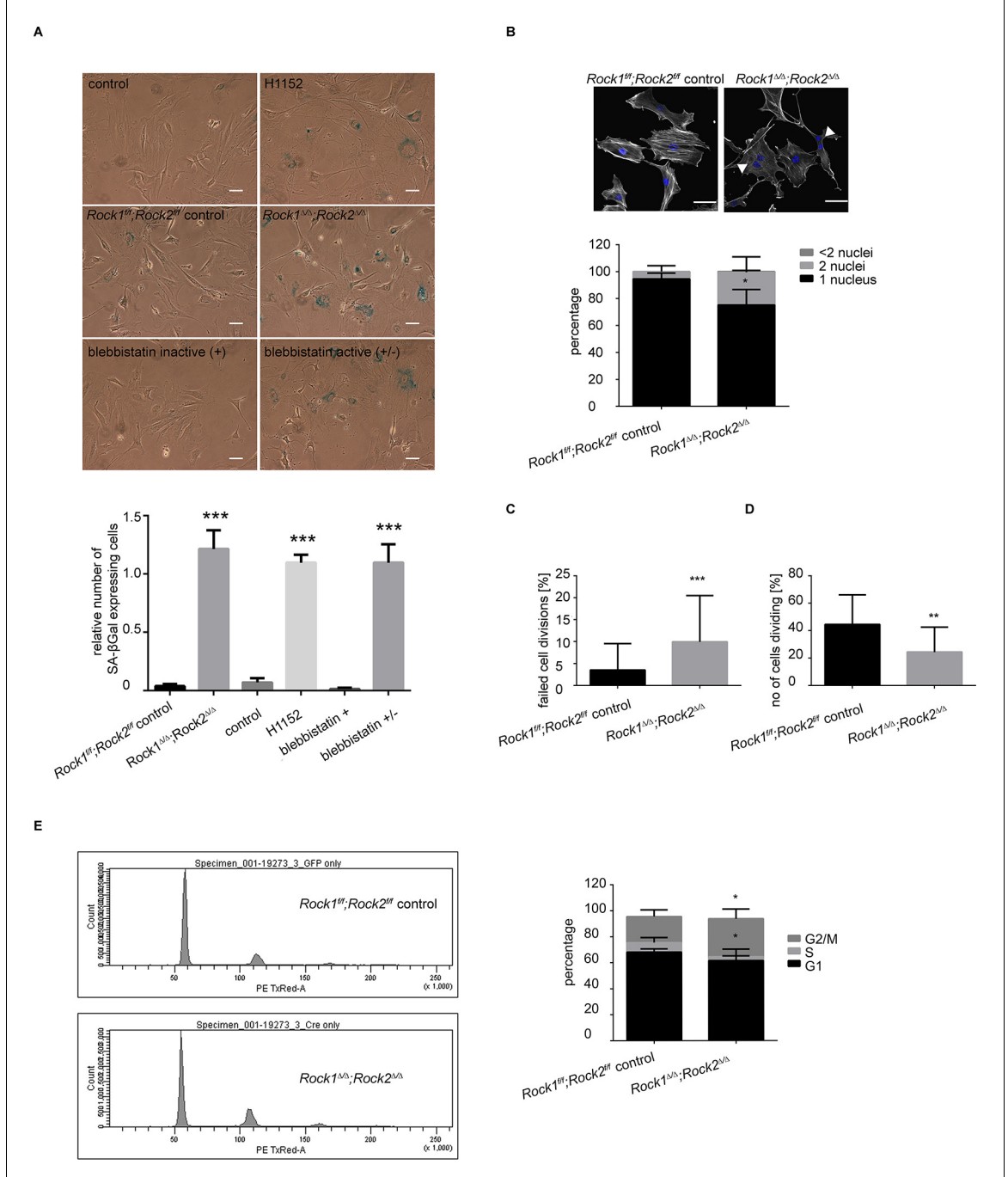

**Figure 3.** *Rock1^{Δ/Δ};Rock2^{Δ/Δ}* cells undergo senescence and show a cell cycle block. (**A**) Images of MEFs treated with H1152, blebbistatin (+), blebbistatin (+/-) for 5 days and *Rock1^{f/f};Rock2^{f/f}* control, *Rock1^{Δ/Δ};Rock2^{Δ/Δ}* MEFs, 5 days after Ad-GFP and Ad-Cre-GFP infection and followed by SA-β gal staining. Scale bars are 50 μm. Graph shows number of SA-βgal expressing cells divided by total number of cells and SD of >100 cells from three independent experiments and p-values were calculated using Student's t-test: *** p<0.001. (**B**) Images of *Rock1^{f/f};Rock2^{f/f}* and *Rock1^{Δ/Δ};Rock2^{Δ/Δ}* MEFs stained with phalloidin and DAPI. Overlay images shown. Arrows in image indicate binucleate cells. Scale bars are 50 μm. Bar chart shows average data and SD of nuclei in > 150 cells from 5 independent experiments. Values were calculated using Student's t-test: * p<0.05. (**C, D**) Analysis of cell division of *Rock1^{f/f};Rock2^{f/f}* and *Rock1^{Δ/Δ};Rock2^{Δ/Δ}* MEFs in time-lapse movies. (**C**) Quantification of failed cell divisions resulting in binucleate cells. (**D**) Quantification of average number of cells dividing. Graphs show average data and SD of > 300 cells from at least five independent experiments. (**E**) Cell cycle profiles of *Rock1^{f/f};Rock2^{f/f}* and *Rock1^{Δ/Δ};Rock2^{Δ/Δ}* MEFs. The graph shows the percentage of cells in G2/M (top), S (middle) and G1 (bottom) phase of the cell cycle. Error bars represent SD. Data are from 5 independent experiments and p-values were calculated using Student's t-test: * p<0.05.

*Figure 3 continued on next page*

*Figure 3 continued*

The following figure supplement is available for figure 3:

**Figure supplement 1.** Analysis of senescence in ROCK-inhibited cells and induction of a cell cycle block in ROCK-depleted cells.

retinoblastoma (Rb) proteins, thereby blocking E2F-dependent transcription (*Brenner et al., 1998*; *Nevins, 2001*). To determine whether blockade of Rb or Trp53 function would overcome senescence resulting from abrogation of ROCK function, we stably infected $Rock1^{f/f};Rock2^{f/f}$MEFs with simian virus 40 large T-antigen (SV40 Large T), which is known to inactivate the tumor suppressors Trp53 and Rb (*DeCaprio et al., 1988*; *Kierstead and Tevethia, 1993*). The cells were then infected with Ad-Cre, to generate $Rock1^{\Delta/\Delta};Rock2^{\Delta/\Delta}$ cells (*Figure 4—figure supplement 1A*). Inactivation of Trp53 and Rb did not prevent the senescence phenotype caused by loss of ROCK, and cells displayed a similar growth defect to the one observed in $Rock1^{\Delta/\Delta};Rock2^{\Delta/\Delta}$ MEFs without SV40 Large T (*Figure 4—figure supplement 1A*). To further confirm that the senescent phenotype did not involve activation of Trp53, MEFs were isolated from $Rock1^{f/f};Rock2^{f/f};Trp53^{f/f}$ mice and ROCK and Trp53 depleted in vitro by Ad-Cre infection. $Rock1^{\Delta/\Delta};Rock2^{\Delta/\Delta};Trp53^{\Delta/\Delta}$ MEFs were found to undergo senescence, and were defective in their proliferation (*Figure 4—figure supplement 1B*). Western blot analysis confirmed depletion of both ROCKs and Trp53 (*Figure 4—figure supplement 1B*). We then treated *Trp53* wild type and null cells with the ROCK inhibitors H1152 and GSK269962A and found an increase in senescence compared to control cells (*Figure 4—figure supplement 1C*). Therefore, the classical senescence pathways did not seem to be affected in $Rock1^{\Delta/\Delta};Rock2^{\Delta/\Delta}$ cells or cells treated for prolonged periods with H1152. To identify protein changes associated with senescence, we used immunoblotting and quantitative mass spectrometry (*Ong et al., 2002*). The protein levels of some cell cycle regulators appeared to change upon treatment with Cre recombinase alone, so for proteomics analysis we focused on long-term H1152 treatment. Annotation enrichment analysis (*Cox and Mann, 2008*) of the proteome-wide changes after long-term ROCK inhibition revealed many cell cycle-related protein categories that were significantly modulated in ROCK-inhibited cells (*Supplementary file 2*), consistent with a prominent cell-cycle phenotype. However, when we analyzed proteins that are known to control G1 cell-cycle arrest and senescence, we did not detect significant changes in the levels of these proteins. These included known regulators, such as p16, p19, p27, p21, CyclinD1, CyclinE, or phosphorylation of Rb (*Supplementary file 2* and *Figure 4—figure supplement 2A,B*). However, CDK1, CyclinA and CKS1 protein levels were significantly reduced upon loss of and long-term inhibition of ROCK1 and 2 (*Supplementary file 2* and *Figure 4A–C*). Long-term treatment of cells with blebbistatin (blebbistatin +/-), but not the inactive enantiomer (blebbistatin +), also decreased the levels of CDK1, CyclinA and CKS1 (*Figure 4B,C*), suggesting that it is loss of contractility downstream of ROCK loss that results in down-regulation of these proteins. However, it cannot be ruled out that both ROCK inhibition/depletion as well as myosin II inhibition independently lead to a similar phenotype.

Cyclin-dependent kinase regulatory subunits 1 and 2 (*Cks1b* and *Cks2*) form a complex with a subset of cyclin-dependent kinases (Cdks) that are the essential regulators of cell cycle progression. Mammalian cells in which *Cks* is silenced exhibit a slowed G1 progression and arrest in G2/M phase of the cell cycle, and are therefore unable to enter mitosis (*Martinsson-Ahlzen et al., 2008*; *Westbrook et al., 2007*). It was previously reported that siRNA knockdown of CKS1 protein in $Cks2^{-/-}Cks1b^{+/-}$ MEFs leads to a decrease in proliferation through defects in *Cdk1*, *Ccna2* and *Ccnb1* transcription and expression of the equivalent proteins CDK1, CyclinA and CyclinB (*Martinsson-Ahlzen et al., 2008*). Analysis of mRNA levels revealed that, in $Rock1^{\Delta/\Delta}/Rock2^{\Delta/\Delta}$ cells, *Cdk1* and *Ccna2* mRNA levels were significantly reduced, whereas *Cks1b* mRNA remained unchanged (*Figure 4D*). APC/C$^{Cdh1}$ (anaphase-promoting complex and its activator Cdh1) have been previously implicated in regulating CKS1 as well as SKP2 stability (*Bashir et al., 2004*). We were unable to detect SKP2 protein in the cells, but CKS1 protein was still downregulated when *Cdh1* null MEFs were treated with H1152 or blebbistatin (data not shown). Furthermore, MG132 treatment of cells did not prevent downregulation of CKS1 suggesting the proteasome is not responsible for this decrease (data not shown).

To determine whether the down-regulated cell cycle regulators are responsible for the phenotype induced by the loss of ROCK 1 and 2, we employed shRNA knockdown of *Cks1b*, *Cdk1* and *Ccna2*.

Cells depleted of CKS1, CDK1 or CyclinA were unable to proliferate (*Figure 5A*) and had increased staining for SA-βGal (*Figure 5B*). Interestingly, it has been shown previously that the knockout of *Cks1b* (*Hoellein et al., 2012*) or *Cdk1* (*Diril et al., 2012*) leads to cellular senescence. We found that knockdown of *Cks1b* affected protein levels of CDK1 and CyclinA, while knockdown of *Cdk1* affected protein levels of CKS1 and CyclinA and knockdown of *Ccna2* affected levels of CKS1 and CDK1 (*Figure 5C*), confirming a clear link between the expression of these proteins. Overall, these results indicate that cell cycle proteins CKS1, CDK1 and CyclinA are downregulated upon loss of actomyosin contractility, and their depletion causes defects in cell proliferation and increased cellular senescence.

## ROCK function is essential for tumorigenesis, but ROCK1 and 2 proteins act redundantly

After thorough in vitro characterization of cells lacking ROCK1 and ROCK2, we investigated the roles of ROCK1 and ROCK2 in tumorigenesis in vivo using genetically engineered mouse tumor models. To this end, we interbred mice carrying the conditional *Rock* alleles with models of lung tumorigenesis and melanoma. Activation of the LSL-oncogenes $Kras^{G12D}$ and $Braf^{V600E}$ in these models by Cre recombinase also leads to deletion of conditional *Rock* alleles.

For lung tumorigenesis, we used a NSCLC model, exploiting activation of $LSL\text{-}Kras^{G12D}$ and $LSL\text{-}Trp53^{R270H}$ in lung epithelial cells (*DuPage et al., 2009*; *Jackson et al., 2005*; *2001*). Mice were treated with adenovirus expressing Cre (Ad-Cre) and sacrificed 24 weeks after treatment. Histological analysis revealed that the total tumor area was increased in $Rock1^{f/f}$ mice (*Figure 6A*). This was seen for all tumor grades (*Figure 6A*) that have been classified as reported previously (*DuPage et al., 2009*; *Jackson et al., 2005*; *2001*). This result suggests that ROCK1 may have a tumor suppressor function in this specific mouse model, of which the mechanism requires further investigation and will be the subject of another manuscript. However, no difference was seen comparing tumor area of $Rock1^{f/f};Rock2^{f/f}$ or $Rock2^{f/f}$ with wild type mice (*Figure 6A*). Western blot analysis of cell lines generated from $Rock1^{f/f};Rock2^{f/f};Kras^{G12D};Trp53^{R270H}$-driven tumors showed that they always retained ROCK1 or ROCK2 protein (*Figure 6B*) and contained un-recombined *floxed* alleles of *Rock1* and *2*. To investigate the consequences of complete loss of ROCK function, we infected these 'retainer' cell lines isolated from lung tumors arising in $Rock1^{f/f};Rock2^{f/f}$ mice, which retained ROCK1 protein ($Rock1^{f/f};Rock2^{\Delta/\Delta}$), with Ad-Cre-GFP to excise the retained *Rock1* locus. The same procedure was applied to cells isolated from tumors arising in $Rock1^{f/wt};Rock2^{f/f}$ mice ($Rock1^{\Delta/wt};Rock2^{\Delta/\Delta}$). $Rock1^{\Delta/\Delta};Rock2^{\Delta/\Delta}$ cells, but not $Rock1^{\Delta/wt};Rock2^{\Delta/\Delta}$ cells, failed to proliferate when compared to the control cells in a colony formation assay (*Figure 6C*). This further highlights the fact that one wild-type allele of *Rock* is sufficient to allow cell proliferation. Importantly, we find that just one allele of either *Rock* is sufficient to rescue the proliferation defect which was also observed using MEFs (data not shown) and this $Rock1^{\Delta/wt};Rock2^{\Delta/\Delta}$ cell line (*Figure 6C*).

In order to quantify ROCK1 and ROCK2 protein abundance from each knockout, we established a custom-selected reaction monitoring (SRM) targeted mass-spectrometry assay (*Worboys et al., 2014*). This enables the analysis of absolute concentrations of each isoform, allowing us to identify

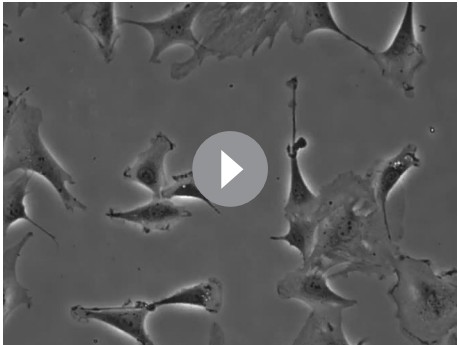

**Video 3.** $Rock1^{f/f};Rock2^{f/f}$ control, 5 days after infection with Ad-GFP, 20X magnification.

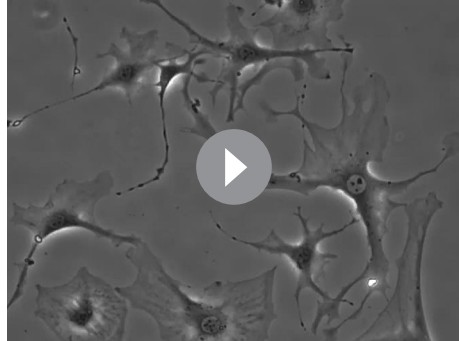

**Video 4.** $Rock1^{\Delta/\Delta};Rock2^{\Delta/\Delta}$, 5 days after infection with Ad-Cre, 20X magnification.

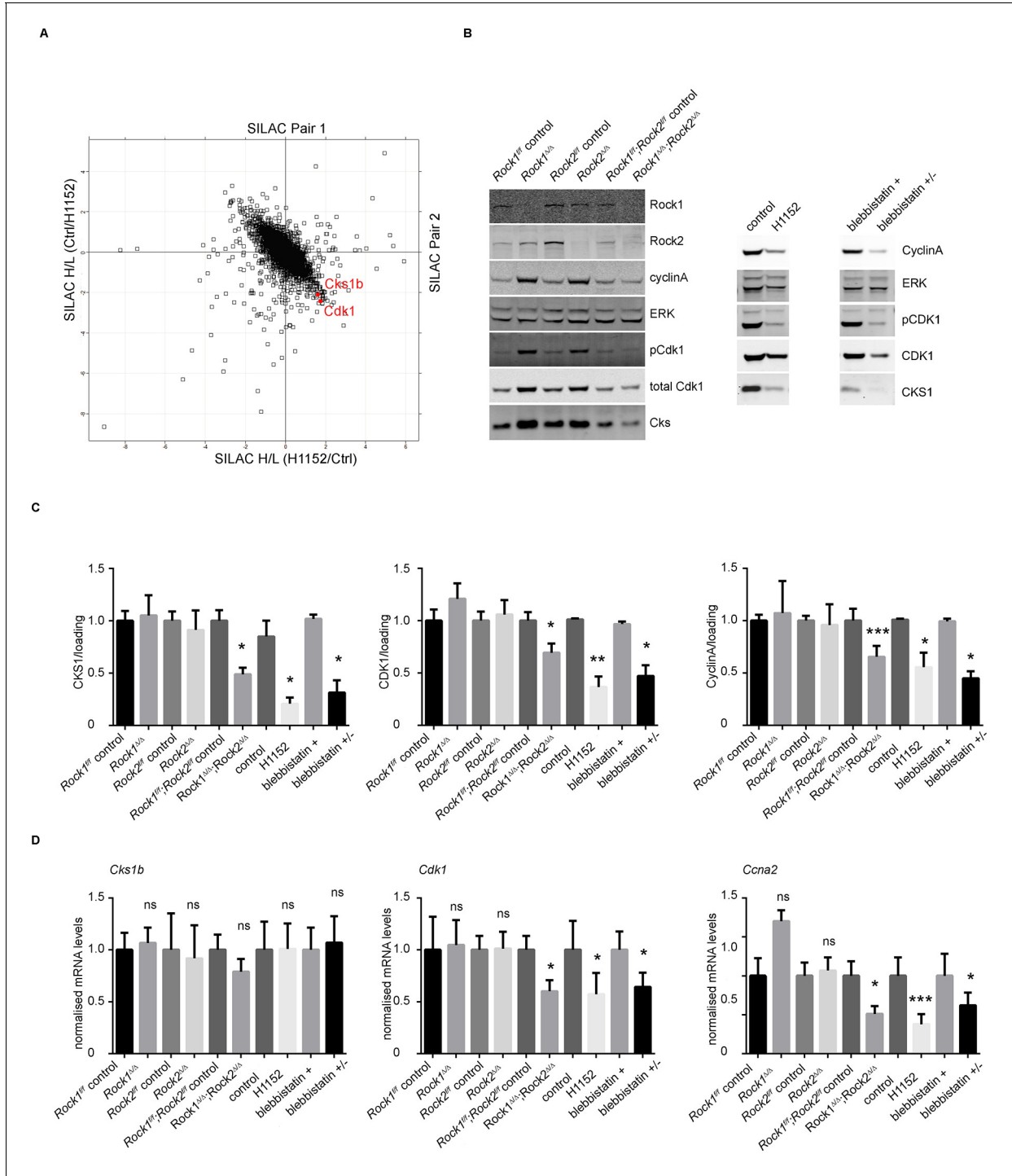

**Figure 4.** Downregulation of CKS1, CyclinA and CDK1 on abrogation of ROCK function. (**A**) MEFs were treated with H1152 or vehicle for 3 days and the proteome analyzed by quantitative mass spectrometry. Graph shows log ratios of identified proteins. Data are from duplicate experiments of reciprocal SILAC labelling. The 'Significant-B' outlier test was used to determine significantly regulated peptides or proteins, using a Benjamini-Hochberg FDR rate of 5%. (**B, C**) MEFs of different genotypes or treated with indicated inhibitors were immunoblotted for pCDK1, total CDK1, Cyclin A, CKS1, ROCK1, ROCK2 and total ERK as loading control (**B**). (**C**) Quantification of western blot analyses. Graphs show protein levels of CDK1, CyclinA and CKS1 divided by total ERK protein levels and SEM in indicated samples. The data are from 5 independent experiments and p-values were calculated using Student's t-test: * p<0.05, ** p<0.005, *** p<0.001. (**D**) qPCR analysis of mRNA levels of *Cks1b*, *Cdk1* and *Ccna2* in indicated samples. Graphs show average normalized mRNA levels and SD from at least five independent experiments, each carried out in triplicates. p-values were calculated using Student's t-test: * p<0.05, *** p<0.001.

*Figure 4 continued on next page*

*Figure 4 continued*

The following figure supplements are available for figure 4:

**Figure supplement 1.** Analysis of cellular senescence in $Rock1^{\Delta/\Delta};Rock2^{\Delta/\Delta};Trp53^{\Delta/\Delta}$ cells and in ROCK-inhibited *Trp53* wild type and *Trp53* null cells.

**Figure supplement 2.** Protein levels of cell cycle regulators in ROCK-inhibited and $Rock1^{\Delta/\Delta};Rock2^{\Delta/\Delta}$ cells.

whether one isoform is more expressed than the other and thus could play a more important role. It will also allow analysis of whether loss of one isoform would lead to a compensatory increase in the levels of the other isoform. Isotopically 'heavy' synthetic quantotypic ROCK1 and ROCK peptides were correlated with ROCK1 and ROCK2 to calculate absolute protein abundance from each sample. SRM analysis showed that, in cell lines derived from tumors in $Rock1^{f/f};Rock2^{f/f}$ mice, either ROCK1 or ROCK2 protein was always retained (*Figure 6D*). ROCK1 and ROCK2 protein was absent in cell lines derived from $Rock1^{f/f}$ or $Rock2^{f/f}$ tumor bearing mice, respectively (*Figure 6D*). Micro-

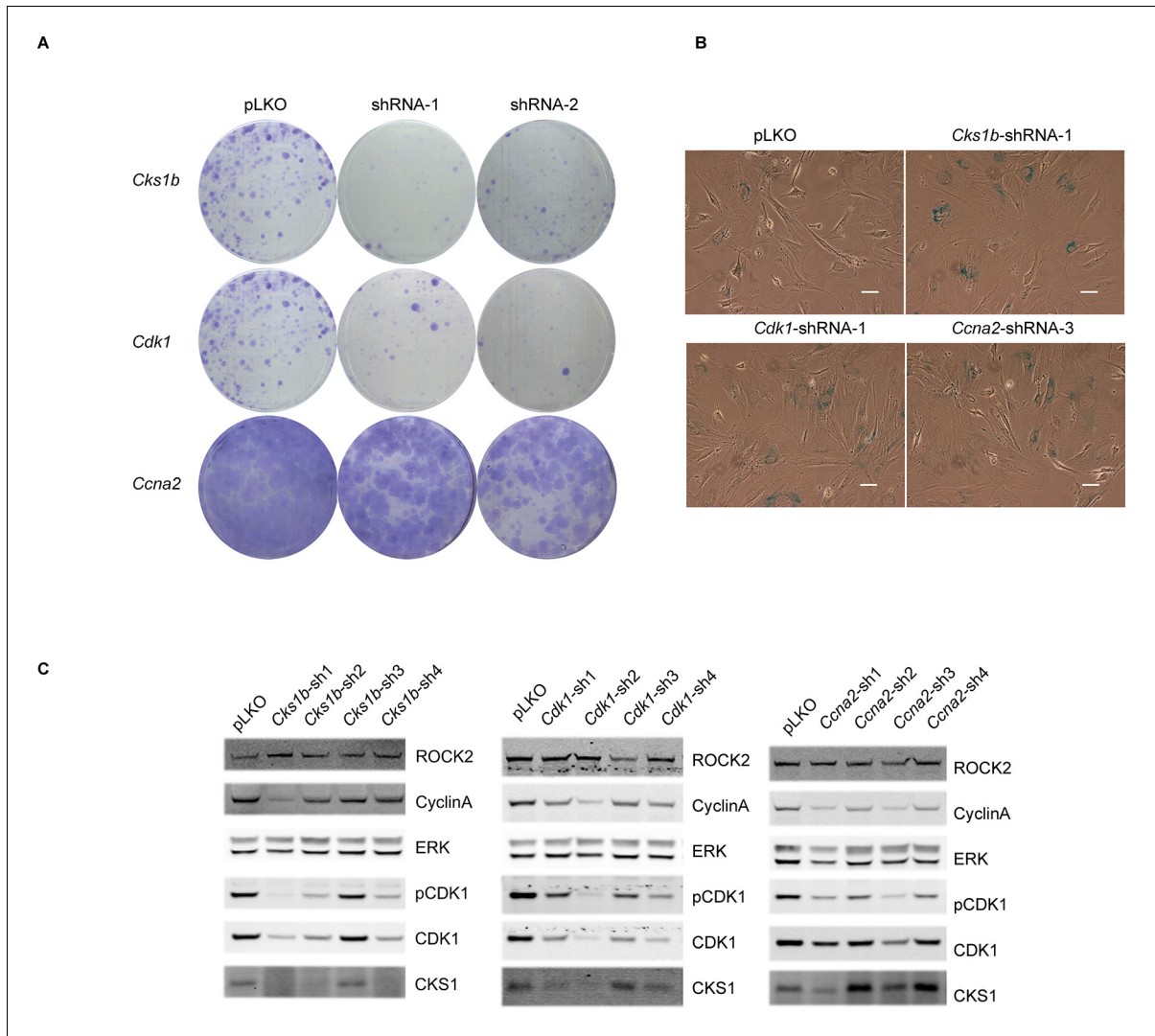

**Figure 5.** Knockdown of *Cdk1* and *Cks1b* leads to cellular senescence. (A–C) MEFs were infected with either control shRNA or shRNAs targeting *Cks1b*, *Cdk1* and *Ccna2*. Four days after infection, equal numbers of cells were plated and kept for seven days, allowing colonies to form (A). Five days after infection with shRNAs, cells were also subjected to SA-βGal staining (B) and immunoblotting with antibodies against pCDK1, total CDK1, Cyclin A, CKS1. ROCK2 and total ERK were used as loading control (C). Scale bars are 50 μm.

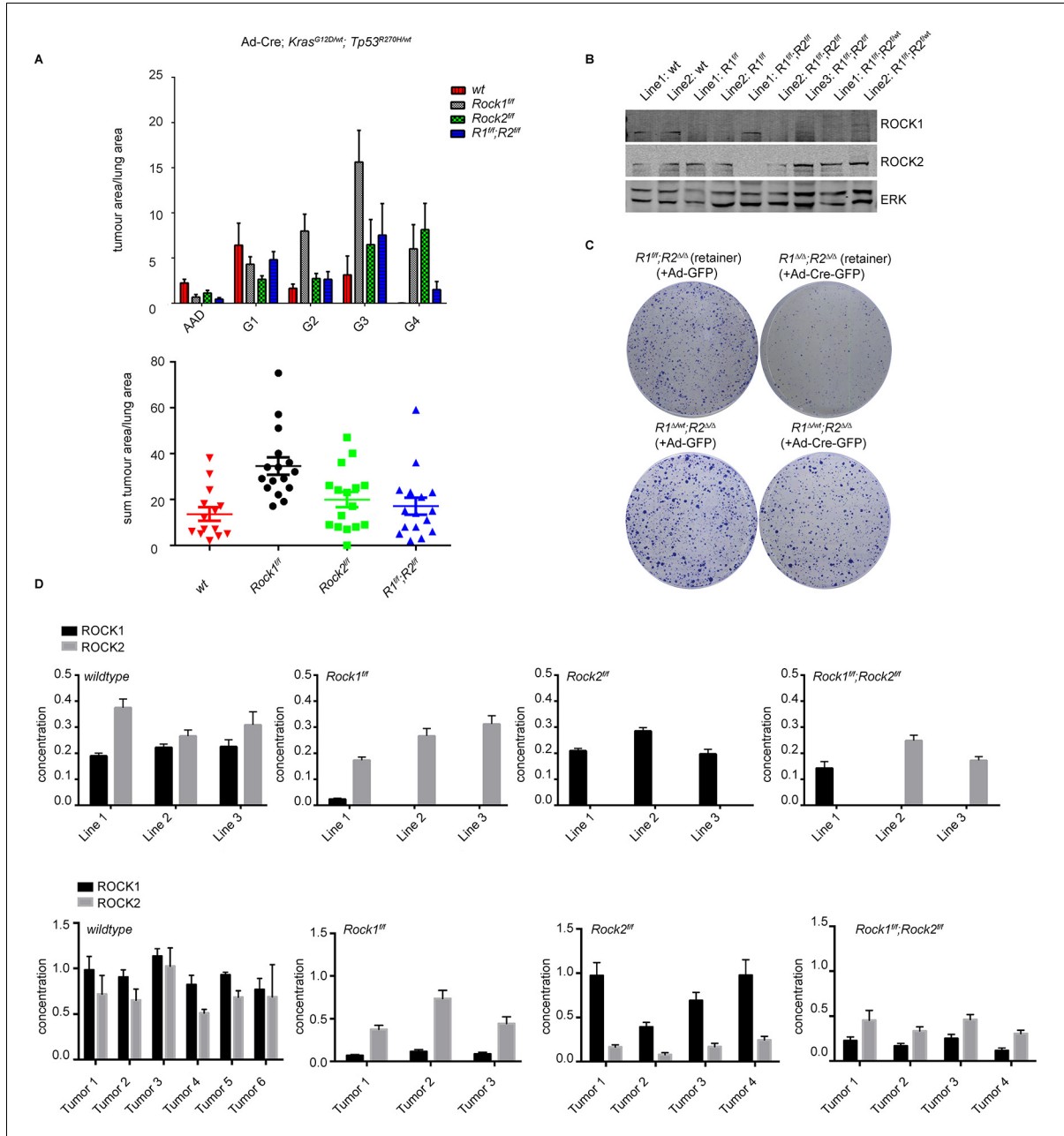

**Figure 6.** ROCK is essential for tumorigenesis. Cohorts of mice carrying *LSL-Kras*^G12D; *LSL-Trp53*^R270H alleles and the indicated *Rock* alleles were treated once with Ad-Cre viruses by intranasal inhalation. 24 weeks later, the mice were sacrificed and the lungs were analyzed by histopathology. (A) Graphs show tumor area divided by lung area for indicated genotypes and tumor grades. The data are also displayed as a sum of tumor area divided by lung area for the indicated genotypes. Graphs show average data and SEM. The total number of mice analyzed is indicated in Scatter Plots. (B) Cell lines derived from *Kras*^G12D;*Trp53*^R270H tumors of either *Rock1*^f/f, *Rock2*^f/f, *Rock1*^f/f;*Rock2*^f/f or wild-type genotype were subjected to immunoblotting with indicated antibodies. (C) Isolated lung tumor cell lines of *Rock1*^Δ/wt;*Rock2*^Δ/Δ genotype as well as a cell line which retained both *Rock1* alleles (*Rock1*^f/f;*Rock2*^Δ/Δ) were treated with Ad-GFP and Ad-Cre-GFP. 3 days after infection, cells were plated and kept for ~5 days allowing colonies to form. (D) Absolute quantification of ROCK1 and 2, using selective reaction monitoring (SRM). Bar graphs show the mean concentration in nM of ROCK1 (black) and ROCK2 (grey) across at least 3 biological replicates from each genetic background. Line 1–3 indicate different cell lines isolated from NSCLCs and Tumors 1–4 represent different micro-dissected tumors. Error bars represent SEM.

The following figure supplement is available for figure 6:

**Figure supplement 1.** Analysis of melanomagenesis in wt, *Rock1*^f/f, *Rock2*^f/f and *Rock1*^f/f;*Rock2*^f/f mice.

dissected tumors from *Rock1^f/f* mice showed a reduction of ROCK1 protein levels but not complete depletion as seen in cell lines. This is likely due to contamination from the wild type stromal cells infiltrating the tumor. The same was also observed in tumors from *Rock2^f/f* mice (*Figure 6D*). In tumors analyzed from *Rock1^f/f;Rock2^f/f* mice, ROCK2 protein was retained but low levels of ROCK1 were also present (*Figure 6D*), most probably from stromal cells. Importantly, ROCK1 protein was not up-regulated in cell lines or tumors from mice with *Rock2^f/f* genotype in a compensatory manner and vice versa. The overall concentration of ROCK1 and 2 proteins was found to be similar, excluding potential effects driven by differential isoform expression.

Similar results were also observed in mouse models of melanoma (*Dhomen et al., 2009*). In the melanoma model, expression of *Braf^V600E* from an LSL-*Braf^V600E* allele is driven in melanocytes by a tamoxifen-activated version of Cre recombinase under the control of the melanocyte-specific tyrosinase promoter (*Tyr-Cre^ERT2*) (*Dhomen et al., 2009*). We also used conditional melanocyte-specific expression of oncogenic *Braf* in combination with a conditional allele of the *Pten* tumor suppressor to decrease tumor latency (*Dankort et al., 2009*; *Dhomen et al., 2009*) (*Figure 6—figure supplement 1A–B*). No differences were found in onset of tumorigenesis (*Figure 6—figure supplement 1A*) or survival (*Figure 6—figure supplement 1B*) when comparing *Braf^V600E* with *Braf^V600E;Rock1^f/f* or *Braf^V600E;Rock1^f/f;Rock2^f/f* mice. Tumor latency was decreased in *Braf^V600E;Rock2^f/f* mice only but not in combination with *Pten^f/f* or *Pten^f/wt* (*Figure 6—figure supplement 1A*). Survival was also found to be decreased in *Braf^V600E:Rock2^f/f* mice as well as in combination with *Pten^f/f* (*Figure 6—figure supplement 1B*), suggesting that unlike in the NSCLC model, ROCK2 may be having a tumor suppressor role in this model.

Nevertheless, similar to the NSCLC model, analysis of cell lines generated from these melanomas, by SRM and immunoblotting, revealed retention of either ROCK protein in mice with *Rock1^f/f;Rock2^f/f* genotype, but efficient excision of the floxed locus in mice with *Rock1^f/f* and *Rock2^f/f* genotypes (*Figure 6—figure supplement 1C–D*). These results further support our observation that ROCK protein is required for tumorigenesis, since retention of at least one *Rock* allele is selected for in both Ras-driven lung tumors and Raf-driven melanomas. Furthermore, treatment of melanoma cell lines with Ad-Cre-GFP, to excise the retained *Rock* locus, revealed that *Rock1^Δ/Δ;Rock2^Δ/Δ* cells but not *Rock2^Δ/Δ* cells fail to proliferate compared to control cells in a colony formation assay (*Figure 6—figure supplement 1E*). This was also seen when compared to *Rock1^Δ/Δ* cells (data not shown).

Collectively, analysis of ROCK1 and ROCK2 protein expression in tumors and tumor derived cell lines suggests that the absence of ROCK1 and ROCK2 is incompatible with tumorigenesis. However, either ROCK1 or ROCK2 protein is sufficient to promote tumorigenesis.

## Discussion

Rho-kinase generates actomyosin contractility and thus has been described as an important regulator of cell motility and plasticity (*Sanz-Moreno et al., 2008*). Several reports using siRNA to knock-down ROCK1 or 2 have been published suggesting that ROCK1 and 2 proteins are not redundant. These include different effects on actomyosin contractility, stress fibers and focal adhesions in fibroblasts (*Yoneda et al., 2005*), opposing effects on cell proliferation and changes in cell migration in glioblastoma cell lines (*Mertsch and Thanos, 2014*). Using ROCK1 and 2 single knockout MEFs, differences in stress fiber formation and Cofilin phosphorylation have been described (*Shi et al., 2013*). In a further study, knockdown of either ROCK1 or ROCK2 alone was sufficient to reduce anchorage-independent growth of NSCLC cell lines - also suggesting their functions are not redundant (*Vigil et al., 2012*). All of these studies, however, describe very different effects of ROCK1 or ROCK2, and at least partially contradict each other. The use of siRNA knockdown, ROCK inhibitors or germ-line knockout MEF lines has several limitations, such as off-target effects, differences in growth properties or cell morphology. To establish an isogenic system and to circumvent the severity of the embryonic phenotypes in germ-line knockouts of *Rock1* and *2* (*Shimizu et al., 2005*; *Thumkeo et al., 2003*; *Thumkeo et al., 2005*), we generated conditional targeted alleles of *Rock1* and *Rock2*, allowing us to analyze of the role of ROCK1 and ROCK2, and permitting the generation of cells that lack both ROCK1 and 2. Our data with genetic deletion in an isogenic fashion show that either ROCK1 or ROCK2 is sufficient to regulate MLC and MYPT phosphorylation and thus actomyosin contractility. Known targets such as the phosphorylation of cofilin are not affected in single or double knockout cells. However, using quantitative mass spectrometry treatment of cells with the

ROCK inhibitor H1152, we show that Cofilin phosphorylation is affected after short-term ROCK inhibition but recovers upon long-term treatment. This indicates that other kinases are able to take over and phosphorylate Cofilin. MLC phosphorylation, however, is not recovered indicating that ROCK is the major kinase regulating MLC phosphorylation.

We demonstrate that the depletion of ROCK1 and 2 leads to severe defects in cell proliferation, which is not observed in cells lacking either ROCK1 or 2. Defects in cell proliferation are also observed in cells treated long-term with ROCK inhibitors or the MyosinII inhibitor blebbistatin. Previous work with the ROCK inhibitor Y-27632 has led to contradictory results on the requirement for ROCK in cell proliferation. We believe that this may reflect the relatively low potency of Y-27632 as a ROCK inhibitor, since we found that treatment with several other ROCK inhibitors (H1152, GSK269962A, AT13148, GSK429286A and chroman1), which have been shown to be more potent than Y-27632 (*Sadok et al., 2015*), could mimic the effects of genetic deletion of *Rock1* and *Rock2*.

Cell proliferation defects upon deletion of ROCK1 and 2 are due to a G1 arrest and a block in cytokinesis. Interestingly, the cell proliferation defects arising from abrogation of ROCK function result in cellular senescence, both in vitro and in vivo. This cellular senescence does not appear to involve changes in levels of the cell cycle inhibitors p16, p21, p27, but the down-regulation of cell-cycle proteins CDK1, Cyclin A and the cyclin-dependent kinase inhibitor CKS1. CKS1 is known to regulate levels of CDK1 and Cyclin A (*Martinsson-Ahlzen et al., 2008*; *Westbrook et al., 2007*), and can function in G1 and G2 phases of the cell cycle (*Tang and Reed, 1993*). Interestingly, it has been shown that $Cks1b^{-/-}$ cells are defective in proliferation, have reduced proportions of cells in S- and G1-phase, accumulation of cells with G2/M DNA content (*Hoellein et al., 2012*; *Spruck et al., 2001*) and exhibit senescence. We therefore propose that an essential function of ROCK in cell proliferation is to regulate the expression of CKS1. In addition, this ROCK-driven CKS1 down-regulation is a novel marker of senescence.

CKS1 does not appear to be regulated at the mRNA level, thus further studies are required to understand how ROCK function could regulate CKS1 protein. Protein degradation was not mediated by the proteasome or APC/C$^{Cdh1}$ complex (data not shown). APC/C$^{Cdh1}$ was previously shown to regulate the degradation of CKS1 and Skp2 (*Bashir et al., 2004*). CKS1 and Skp2 are the targeting subunits of the SCF$^{Skp2-Cks1}$ ubiquitin ligase regulating p21 and p27 degradation (*Sutterluty et al., 1999*). Even though we were unable to detect Skp2 in our system, we do show that p21 and p27 levels are not affected upon loss of ROCK, suggesting a novel regulation of CKS1 and function in mediating senescence. More work is required to unravel the complete mechanism and determine how downregulation of these proteins is mediated.

We observed that treatment with blebbistatin a direct inhibitor of myosin II led to cell cycle arrest, senescence and reduced levels of CKS1, CyclinA and CDK1, potentially suggesting that ROCK-dependent regulation of cell proliferation is through the inhibition of actomyosin contractility rather than another target of ROCK signaling.

Using tumor samples as well as tumor derived cell lines, from two different genetically engineered mouse models of tumorigenesis, we find that tumor formation is incompatible with homozygous deletion of both *Rock1* and *Rock2,* since tumors arising in $Rock1^{f/f};Rock2^{f/f}$ mice always retained either un-recombined $Rock1^{f/f}$or $Rock2^{f/f}$ alleles and showed expression of either ROCK1 or ROCK2, respectively. Targeting either *Rock1* or *Rock2* appears to have mild and divergent consequences in the mouse models. Deletion of *Rock1* appears to increase tumor burden in the NSCLC model, whereas deletion of *Rock2* leads to decreased tumor onset and survival in the melanoma model. These putative tumor suppressor functions of *Rock1* and *Rock2* do not appear to be a consequence of differential expression as the absolute concentration of the two isoforms was found to be similar by SRM. These results require further investigation and will be the basis of a separate manuscript.

The retention of ROCK protein in $Rock1^{f/f};Rock2^{f/f}$ mice, however, demonstrates that ROCK function is essential for tumorigenesis. Incomplete excision of proteins that affect cell proliferation has been reported in several studies (*Kissil et al., 2007*; *Peschard et al., 2012*). In the tumor models used in this study, excision of *Rock* alleles and tumor initiation are achieved simultaneously in an inducible fashion; therefore we propose that the activation of Cre recombinase leads to the formation of a mixed population of cells in which either *Rock,* or both *Rock1* and *2,* have been excised. The ROCK1 and 2 double-null cells, however, are unable to form a tumor as they are defective in cell proliferation and senesce. The remaining cells, that is those that have retained either *Rock* allele form tumors but these are the genetic equivalent of a single knockout.

In summation, our data reveal that ROCK proteins act redundantly to regulate actomyosin contractility, and that this function of ROCK is necessary, by supporting levels of the cell cycle regulators Cyclin A, CDK1 and CKS1, for senescence, cell cycle progression and proliferation both in vitro and in vivo. Recent studies have identified a role for ROCK inhibition in combinatorial treatments in BRAF-mutant melanoma (*Smit et al., 2014*), as well as in Kras-driven lung cancers (*Kumar et al., 2012*). Our data thus provide a molecular basis, and highlight the need, for exploiting the use of effective ROCK inhibitors as anti-cancer agents either alone or in combination. However, incomplete or specific inhibition of only one isoform of ROCK alone may not be an effective treatment, and may even lead to adverse results as our results also highlight isoform specific tumor suppressive functions for ROCK1 in lung and ROCK2 in melanoma models.

## Materials and methods

### Mice

All animal procedures were approved by the Animal Ethics Committee of the Institute of Cancer Research in accordance with National Home Office regulations under the Animals (Scientific Procedures) Act 1986. *Rock1* and *2* gene-targeting vectors were made by inserting arms of homology into the PGKneoF2L2DTA vector (Phillipe Soriano, Addgene plasmid #13445 [*Hoch and Soriano, 2006*]), where loxP sites flank exon 6 of *Rock1* (*Rock1$^{f/f}$*) and exons 5–6 of *Rock2* (*Rock2$^{f/f}$*). Homologous recombination was performed in Bruce4 ES cells of C57BL/6J origin (Ozgene). Positive clones were identified by PCR and confirmed by Southern blotting before injection into albino C57BL/6J blastocysts to generate chimeras in a pure C57BL/6J background. *Rock1*- and *2*-targeted mice were subsequently crossed to *PGK-Flp* transgenic mice (provided by the Cancer Research UK transgenic facility) to generate conditional knockout mice.

For the lung tumor model *Rock1* and *2* targeted mice were crossed to *LSL-Kras$^{G12D}$* (the mice were a generous gift from David Tuveson [*Tuveson et al., 2004*]) and *Trp53$^{R270H}$*. Mice were anesthetized and treated once by intranasal inhalation using $2 \times 10^6$ pfu/mouse of Adeno-Cre viruses as described previously (*DuPage et al., 2009*). Tumor grading is according to the Jacks' lab classification; AAD: Atypical Adenomatous Dysplasia not forming any solid elements; G1: Uniform small rounded nuclei; G2: Enlarged nuclei with some variation and hyperchromatism, small nucleoli; G3: Markedly enlarged and variable nuclei, prominent nucleoli, occasional mitotic figures; G4: Pleomorphic nuclei, prominent nucleoli, mitotic figures easily found, focal necrosis.

For the melanoma model, *Rock1* and *2* targeted mice were crossed to mice carrying the transgenes *LSL-Braf$^{V600E}$*, *Pten$^{f/f}$* (*Pten$^{f/wt}$*, *Pten$^{wt/wt}$*) and *Tyr-Cre$^{ERT2}$* (the mice were a generous gift from Richard Marais). Briefly, two treatment regimens were used; mice carrying *Pten$^{f/wt}$* and *Pten$^{wt/wt}$* targeted alleles were treated with 100 µl of amoxifen (100 mg/ml in ethanol), tamoxifen was applied evenly, to the shaven skin on the dorsal surface of 8-week-old mice. Mice carrying *Pten$^{f/f}$* targeted alleles were treated with 0.5 µl of 4-Hydroxytamoxifen (4-OHT) (10 mM in DMSO); 4-OHT was applied to shaven skin on the dorsal surface of 6-week-old mice three times on three consecutive days.

### Cell culture, reagents and treatments

MEFs were isolated from E13.5 embryos and cultured in a low-oxygen (3%) incubator (*Parrinello et al., 2003*) in Dulbecco's modified Eagle's medium containing 10% fetal bovine serum. Tumors from mouse models were incubated in trypsin and put into culture to allow cells to grow out. Melanomas were incubated at 20% oxygen for fibroblasts present in the culture to undergo senescence and melanoma cells to grow out. All cell lines were subsequently analyzed by PCR and immunoblotting and their genotypes confirmed. 690cl2 cells were generated in-house by N. Dhomen and R. Marais from tumors arising in the *Braf$^{V600E}$*, *Pten$^{f/f}$* mouse model (*Dhomen et al., 2009*). *Trp53* MEFs were generated in-house from the *Trp53* knockout mouse (*Harvey et al., 1993*) and their genotype confirmed by PCR.

All cell lines used for this study were generated in house and tested free of mycoplasma. Adenoviruses were obtained from the Gene Transfer Vector Core at the University of Iowa. SMARTchoice lentiviral shRNA vectors were obtained from Dharmacon and produced in 293T cells co-transfected with pMD2.G and psPAX2 packaging vectors (http://tronolab.epfl.ch/lentivectors). Where indicated,

cells were infected using amphotropic retrovirus produced by transfection of pBABE vectors into Phoenix packaging cells (https://web.stanford.edu/group/nolan/protocols/pro_optimiz.html). pBabe-puro-RasV12 and pBABE-hygro-Trp53DD were a kind gift from Bob Weinberg (Addgene plasmids # 1768, # 9058) (*Hahn et al., 2002*), pBABE-puro-SV40-LT was a kind gift from Thomas Roberts (Addgene plasmid # 13970) (*Zhao et al., 2003*).

Primary antibodies were used, unless indicated otherwise, at a dilution of 1:1000 for immunoblotting and 1:200 for immunofluorescence. Secondary antibodies were used at a dilution of 1:10,000 for immunoblotting (LI-COR Biosciences) and 1:500 for immunofluorescence (Alexa Fluor, Life Technologies)

The following antibodies were used: p44/42 MAPK (ERK1/2) (1:3000) (Cell Signaling), phospho-p44/42 (ERK1/2) (Thr202/Tyr204) (Cell Signaling), GAPDH (1D4) (Novus Biologicals), Rock1 (H-85) (Santa Cruz Biotechnology), Rock2 (ROKα) (BD Biosciences), phospho-MLC (Thr18/Ser19) (Cell Signaling), phospho-MLC (Ser19) for immunofluorescence (Cell Signaling), MRCL3/MRLC2/MYL9 clone (E-4) (Santa Cruz Biotechnology), p21/CIP1 (M-19) (Santa Cruz Biotechnology), Cks1 (Invitrogen), p34cdc2 (Invitrogen), Cdc2 (Y15) (Cell Signaling), CyclinA (CY-A1) (Sigma Aldrich)

Inhibitors used were H1152 (Calbiochem/Merck Millipore), GSK429286A (Selleckchem), GSK269962A (Axon Medchem), chroman1 (ApexBio), Blebbistatin + and Blebbistatin +/- (Calbiochem/Merck Millipore), nocodazole (Sigma Aldrich), AT13148 was synthesized in-house (*Yap et al., 2012b*).

## Immunofluorescence and microscopy

Cells were seeded on glass coverslips, or inside a 1.8 mg/ml collagen gel layer prepared according to manufacturer's instructions (PureCol, Advanced Biomatrix), and left to set in 1 μ-slide chambers (ibidi). The cells were then fixed with 4% paraformaldehyde, permeabilized with 0.1% Triton X-100 and blocked with 2% BSA before staining with phospho-MLC (Ser19) (Cell Signaling) and F-actin (Phalloidin; Life Technologies) and mounting in Vectashield Hardset mounting medium containing DAPI. Confocal images were acquired using a Zeiss LSM 710 microscope (40x oil immersion objective).

Time-lapse phase-contrast microscopy was performed in a humidified $CO_2$ chamber under a Diaphot inverted microscope (Nikon, UK) with a motorized stage (Prior Scientific, UK) controlled by Simple PCI software (Compix). Cells were monitored by taking an image every 10 min over a period of 16–24 hr and their division quantified. For analysis of cell motility, cells were tracked using ImageJ analysis software (http://rsb.info.nih.gov/ij) and the ibidi chemotaxis and migration tool (www.ibidi.com) used to determine migration speed (in μm/min) and directionality (displacement/track length, where displacement is the distance from the start to end point for each cell) were determined using ImageJ analysis software (http://rsb.info.nih.gov/ij) and ibidi chemotaxis and migration tool (www.ibidi.com).

## Immunoblot analysis

Cells were grown to approximately 80% confluence, washed once in PBS and lysed in SDS loading buffer (100 mM Tris pH 6.8, 10% Glycerol, 2% SDS, [100 mM DTT]), homogenized by sonication for 5 s and boiled for 10 min. DTT, to a final concentration of 100 mM, was added after measuring protein concentration by Pierce BCA Assay (Thermo Scientific) and before boiling. Cell lysates containing equal amounts of protein were resolved by SDS-PAGE on precast NuPAGE Novex 10% Bis-Tris Midi Protein Gels (Life Technologies) and transferred onto PVDF-FL membranes (Millipore). Membranes were incubated with indicated antibodies and visualized by fluorescently conjugated secondary antibodies using the Odyssey Infrared Imaging System (LI-COR Biotechnology, Cambridge, UK).

## Gel contraction assay

A total of $1\times10^6$ cells were embedded in 500 μl of a bovine Collagen I mix prepared according to the manufacturer's protocol (PureCol, Advanced Biomatrix) with a final concentration of approximately 2.1 mg/ml in a 24-well plate format. The gel was incubated at 37°C for 1 hr for it to polymerize and 500 μl of medium, containing inhibitors where indicated, was added. A pipette tip was then used to loosen the gel from the tissue culture vessel and incubated for 16 hr. The plates were scanned and percentage contraction was calculated by using the formula 100-([area of the gel/area

of an empty well]*100) of which areas were determined using ImageJ software (http://rsbweb.nih.gov/ij/) (*Hooper et al., 2010*).

## Flow cytometry

Cells were fixed in ice-cold 70% ethanol for at least 1 hr at 4°C and washed twice with PBS. DNA was stained with propidium iodide (PI) (Sigma Aldrich) at a concentration of 20 μg/ml and in the presence of 100 μg/ml RNaseA (Qiagen). The samples were analyzed on a BD LSR II flow cytometer using BD FACSDiva software (BD Biosciences). Where indicated, cells were treated with Nocodazole.

## SA-βgal assay in vitro and in vivo

Staining to detect senescence-associated beta-galactosidase (SA-βgal) activity was carried out as described by Debacq-Chainiaux et *al.* (*Debacq-Chainiaux et al., 2009*). For in vivo staining, the mouse melanoma cell line (690cl2) used was generated from tumors arising in mice carrying $Braf^{V600E}$, $Pten^{f/f}$, $Tyr-Cre^{ERT2}$ after induction by Tamoxifen treatment (*Dhomen et al., 2009*). $2x10^5$ melanoma cells were injected intra-dermally into the lateral flanks of 6–8 week-old CD1 athymic mice; two days after injection the mice were randomly divided into two groups (three mice in each group), one group was treated with vehicle only and the other with the drug. AT13148 was administered at 40 mg/kg in 10% DMSO, 1% Tween-20 in saline, p.o., the dosing schedule used was 2/5 days. At the end of the experiment, the tumors were removed, snap-frozen and senescence-associated beta-galactosidase activity assessed on frozen sections using the protocol described above.

## Mass spectrometry

SILAC heavy and light labelled MEFs were treated for indicated times with H1152, or vehicle, in duplicate (heavy vs. light as well as light vs. heavy). Cells were then lysed by 2% SDS, Tris-HCl pH 7.5, and protein concentrations were measured and balanced by BCA assay. The reciprocal labels of drug and vehicle treated lysates were then mixed 1:1, resulting in two reciprocally labelled SILAC replicates (Heavy drug + light vehicle and vice versa). The mixed lysates were then reduced by addition of 100 mM DTT and boiling for 5 min, followed by trypsin digestion using a Filter Aided Sample Preparation (FASP) protocol (*Wisniewski et al., 2009*). For phospho-peptide analysis, the digests were subjected to phospho-peptide enrichment on TiO2 tips (GL sciences). For total proteome analysis, the digests were initially fractionated by isoelectric focusing, using an OFF-gel fractionator (Agilent, pH 3–10). The fractions were then cleaned up using C18 ziptip columns (Millipore). All samples were analyzed by LC-MS/MS using a Thermo Orbitrap-Velos mass spectrometer (ICR proteomics core facility). The raw data wereas then searched and quantified using Maxquant (*Cox and Mann, 2008*). All subsequent statistical analysis was performed using Perseus software from the Maxquant package. Briefly, the ratios were converted to Log-2 scale, and reverse and contaminants were filtered out. Proteins only identified in one replicate were also filtered out. The 'Significant-B' outlier test was used to determine the significantly regulated peptides or proteins, using a Benjamini-Hochberg FDR rate of 5%. A hit was considered significant if it was found to be significantly changing in opposite directions in both reciprocal SILAC replicates. 2-Dimensional Annotation enrichment analysis was also performed in Perseus, using Broad Institute's Gene Set Enrichment Analysis (GSEA) annotations, using a Benjamini-Hochberg FDR rate of 2%. All proteomics data graphs were generated in Perseus. The targeted proteomics dataset has been deposited to PRIDE: *Figure 4A* accession PXD002521, *Figure 2—figure supplement 1B–C* accession PXD002515.

## Quantitative PCR

RNA was isolated using the RNeasy Mini Kit (Qiagen), according to manufaturer's instructions, and QuantiTect Primer Assays, purchased from Qiagen, were used. The Brilliant II SYBR Green QRT-PCR 1-Step system was used according to the manufacturer's instructions (Agilent Technologies). The reactions were analysed on an ABI PRISM 7900HT Sequence Detection System (Applied Biosystems).

### Absolute protein level quantitation by selected reaction monitoring

A previously reported approach for absolute quantitation of proteins (AQUA) based on standard stable isotope labelled peptides was used (*Gerber et al., 2003*). 'Heavy' isotopic peptides were purchased from Thermo Scientific (USA). SRM was performed, as described elsewhere (*Worboys et al., 2014*), for the measurement of endogenous and synthetic peptides. To select optimal proteotypic peptides to monitor by SRM, all possible proteotypic peptides of ROCK1 and 2 were empirically measured within cell lines that were *wt*, *Rock1*$^{-/-}$ and *Rock2*$^{-/-}$, permitting a confirmation of peptide specificity. For each SRM analysis bands from an SDS-PAGE gel were excised between 150–250 kDa (Precision Plus Protein Prestained Standard, Bio-Rad), and proteins trypsin-digested in-gel. NIDNFLSR, LLLQNELK and ESDIEQLR were chosen for ROCK1 and VYYDISSAK, QLDEANALLR and ENLLLSDSPPCR for ROCK2. For quantitation, the two most intense transitions from each peptide were summed for each isotopic variant. A cycle time of 1.8 s was used, and transitions monitored in an unscheduled manner giving a dwell time of 37.5 ms. Standard curves were generated within sample matrices, and plotted as a function of their SILAC ratios. Subsequent regression lines were used per peptide to calculate absolute concentrations. In tumor samples, 5 fmol of each peptide was spiked into each sample. Three technical replicates were performed and averaged per peptide and each average peptide measurement was averaged per protein. Thus, for each protein a total of 9 measurements are made. The targeted proteomics dataset has been deposited to the PeptideAtlas SRM Experiment Library (PASSEL) with the identifier PASS00721.

## Statistical analysis

Statistical tests were conducted with GraphPad Prism version 6 or Microsoft Excel 2010. Unless stated otherwise, p-values were generated using two-tailed, paired Student's t-test and considered significant if $p < 0.05$. * $p < 0.05$; ** $p < 0.01$ *** $p < 0.001$. Error bars represent standard deviation (SD) or standard error of the mean (SEM) as indicated in figure legends.

## Acknowledgements

We thank Anne Ridley and George Damoulakis for helpful comments and corrections of manuscript, Pascal Peschard for help with experiments, reagents and protocols, Georgia Mavria for help with breedings and the ICR Proteomics Core Facility for mass spectrometry runs. We thank Richard Marais for providing mice and cell line 690cl2. We thank Michelle Garrett and Ian Collins for providing the AT13148. Finally, we would like to thank all other Marshall Lab members for their comments and discussions. This work was funded by Cancer Research UK.

## Additional information

### Funding

| Funder | Grant reference number | Author |
| --- | --- | --- |
| Cancer Research UK | C107/A1205 | Sandra Kümper<br>Faraz K Mardakheh<br>Afshan McCarthy<br>Maggie Yeo<br>Amine Sadok<br>Christopher J Marshall |
| Cancer Research UK | C107/A10433 | Sandra Kümper<br>Faraz K Mardakheh<br>Afshan McCarthy<br>Maggie Yeo<br>Angela Paul<br>Amine Sadok<br>Christopher J Marshall |

| Cancer Research UK | C107/A16512 | Sandra Kümper |
| | | Faraz K Mardakheh |
| | | Afshan McCarthy |
| | | Maggie Yeo |
| | | Angela Paul |
| | | Amine Sadok |
| | | Christopher J Marshall |

The funders had no role in study design, data collection and interpretation, or the decision to submit the work for publication.

## Author contributions

SK, Designed all experiments and wrote the manuscript; Performed breeding, treatment and analysis of mouse tumor models; Performed all other experiments and data analyses, Conception and design, Acquisition of data, Analysis and interpretation of data, Drafting or revising the article; FKM, Mass spectrometry analyses, Acquisition of data, Analysis and interpretation of data, Drafting or revising the article; AM, Performed breeding, treatment and analysis of mouse tumor models; Carried out the AT13148 in vivo senescence experiments, Acquisition of data, Analysis and interpretation of data, Drafting or revising the article; MY, Performed quantitative PCR and Western Blots, Acquisition of data, Analysis and interpretation of data; GWS, Histopathology analysis of NSCLC, Acquisition of data, Analysis and interpretation of data; AP, Assisted with mass spectrometry sample prep, Acquisition of data, Analysis and interpretation of data; JW, SRM analysis, Acquisition of data, Analysis and interpretation of data; AS, Carried out the AT13148 in vivo senescence experiments, Conception and design, Acquisition of data; CJ, SRM analysis, Conception and design, Acquisition of data; SG, Generated conditional mice, Acquisition of data, Analysis and interpretation of data; CJM, Designed all experiments and wrote the manuscript, Conception and design, Acquisition of data, Analysis and interpretation of data, Drafting or revising the article

## Author ORCIDs

Sandra Kümper, http://orcid.org/0000-0002-5099-8070
Jonathan Worboys, http://orcid.org/0000-0002-8242-4901

## Ethics

Animal experimentation: All animal procedures were approved by the Animal Ethics Committee of the Institute of Cancer Research in accordance with National Home Office regulations under the Animals (Scientific Procedures) Act 1986. The date of approval of the current project license under which this work was carried out was the 07/09/13.

## Additional files

### Supplementary files

• Supplementary file 1. Identified phosphorylated proteins in MEFs treated with H1152 for 20 min or overnight versus vehicle.

• Supplementary file 2. Identified total proteins upon long-term treatment of cells with H1152 versus vehicle.

### Major datasets

The following datasets were generated:

| Author(s) | Year | Dataset title | Dataset URL | Database, license, and accessibility information |
|---|---|---|---|---|
| Kümper S, Mardakheh FK, McCarthy A, Yeo M, Stamp GW, Paul A, Worboys J, Sadok A, Jørgensen C, Guichard S, Marshall CJ | 2016 | Analysis of total proteome changes upon long-term treatment with Rho kinase inhibitor H1152 | http://www.ebi.ac.uk/pride/archive/projects/PXD002521 | Publicly available at the EBI PRIDE Archive (Accession no: PXD002521). |
| Kümper S, Mardakheh FK, McCarthy A, Yeo M, Stamp GW, Paul A, Worboys J, Sadok A, Jørgensen C, Guichard S, Marshall CJ | 2016 | Temporal analysis of phosphoproteome changes following Rho-kinase | http://www.ebi.ac.uk/pride/archive/projects/PXD002515 | Publicly available at the EBI PRIDE Archive (Accession no: PXD002515). |
| Kümper S, Mardakheh FK, McCarthy A, Yeo M, Stamp GW, Paul A, Worboys J, Sadok A, Jørgensen C, Guichard S, Marshall CJ | 2016 | Rho-associated kinase function is essential for cell cycle progression and tumorigenesis | http://www.peptideatlas.org/PASS/PASS00721 | Publicly available at Peptide Atlas (Accession no: PASS00721). |

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
