## [Decision Letter]

Thank you for submitting your work entitled "Rho-associated kinase (ROCK) regulation of actomyosin contractility is essential for cell cycle progression and senescence" for consideration by *eLife*. Your article has been reviewed by two peer reviewers, one of whom is a member of our Board of Reviewing Editors. The evaluation has been overseen by the Reviewing Editor and Ivan Dikic as the Senior Editor.

The reviewers have discussed the reviews with one another and the Reviewing editor has drafted this decision to help you prepare a revised submission.

This is an important study that describes the effect of Rock-deficiency on proliferation and tumorigenesis. The technical advance described is the use of floxed Rock1/2 mice for these studies, although some studies are performed using Rock inhibitors. The authors conclude that Rock is required for proliferation and that Rock-deficiency causes senescence mediated by a Trp53-independent pathway by down-regulation of Cdk1, CycA, and CKS1. A requirement of Rock for tumor formation was established using mouse models of lung cancer (Ras-driven) and melanoma (B-Raf-driven).

The data presented are convincing, but there are several areas of ambiguity and revision of the manuscript is required to clarify these issues.

1) The authors should explicitly document the time course of events following Rock gene ablation. The authors report that a "tail retraction" phenotype followed by a flattened morphology at later times. What is the time course of Rock protein loss and MLC phosphorylation? Is Rock lost first and the senescence phenotype immediately observed or does the senescence phenotype gradually develop at some time after loss of Rock protein? A time course description of these events would be helpful.

2) In the Introduction, the authors refer to the literature by stating that previous studies on Rock have focused on cell migration. Omission of an analysis of cell migration in this study appears to be a weakness. Clearly, senescent cells will have a migration defect, but can cell migration by examined after Rock protein loss and before the onset of senescence? The manuscript would be improved by addressing this issue.

3) The Trp53-independence of the senescence is not completely convincing based on the data presented. The key experiment requires 6 floxed alleles to be deleted. It is important that the ablation of Trp53 and Rock genes and loss of protein expression are documented in this study to show that Trp53 is not required for senescence.

4) The mechanism of Trp53-independent senescence is not well established – it may require Cdk1, CycA, and CKS1, but the mechanism is unclear and the stated hypothesis is not tested. As an initial description, this is acceptable, but these caveats should be stated. It is also unclear why the text invokes Cdk1, CycA, and CKS1 while the Abstract only refers to Cdk1 and CKS1. Moreover, in subheading “Loss of ROCK-dependent actomyosin contractility leads to cellular senescence”, the authors argue that there is a block in G1, while in the Abstract, the authors indicate that it was a multi-stage cell cycle arrest. The wording is confusing and should be clarified in the Abstract and the text of the Results section of the manuscript.

5) The purpose of the SRM analysis of the tumor cell lines is unclear since the presence of Rock can be detected by western. What is the evidence that the tumor cells are not a mixture of Rock positive and negative cells? It appears that the presence of Rock is interpreted as if all the cells must express Rock – importantly, this interpretation is required for the authors' conclusions. A more rigorous test would be immunostaining of tumor sections with antibodies to Rock and/or phospho Ser-19/Thr-20 MLC if that is technically feasible.

6) Paragraph two, subheading “Depletion of both ROCK1 and 2 leads to multi-stage cell cycle arrest through defects in actomyosin contractility”. To test whether the effect of Rock depletion is caused by actomyosin contractility, the authors examined the effect of blebbistatin. Similar comments are made throughout the manuscript. This experiment is not a formal test of whether the Rock phenotype is caused by a change in actomyosin contractility – it is only a correlation that the phenotypes caused by blebbistatin and Rock depletion are similar. These statements should be re-worded throughout, including the title of the manuscript.

7) Figure 1 does not necessarily show a 'significant slower' growth rate as stated – but rather sluggish growth, perhaps caused by a delay in growth initiation. If these mice had been allowed progress further, the rate of growth might have been restored and the slope indistinguishable. In their mice experiments, was there any evidence for senescence in vivo at the phase of the growth curve where there was a visible difference in growth – perhaps around day 15 in Figure 1?

8) The key blots should be quantified. Due to the different exposures and background, it is a little difficult to assess the degree of protein changes at times.

---

## [Author Response]

*The data presented are convincing, but there are several areas of ambiguity and revision of the manuscript is required to clarify these issues.*

In response to the raised concerns, several new additional supplementary data has been added to the revised manuscript (Figure 2—figure supplement 1; Figure 3—figure supplement 1; Figure 4—figure supplement 1).

*1) The authors should explicitly document the time course of events following Rock gene ablation. The authors report that a "tail retraction" phenotype followed by a flattened morphology at later times. What is the time course of Rock protein loss and MLC phosphorylation? Is Rock lost first and the senescence phenotype immediately observed or does the senescence phenotype gradually develop at some time after loss of Rock protein? A time course description of these events would be helpful.*

We find that ROCK is gradually lost over a time course of 5 days, at which point, it was when it was found to be completely depleted. We have now included a time course showing ROCK depletion after Ad-Cre-GFP and Ad-GFP infection (Figure 2—figure supplement 1). Cells display a tail retraction phenotype 3 days after adenovirus infection when ROCK levels are reduced but not completely depleted. After 5 days levels are depleted to a maximum and cellular senescence is seen. When inhibitors such as H1152 are used to analyze senescence, similar results are observed, starting with an initial tail retraction phenotype and finally cellular senescence after prolonged exposure to the inhibitors. To clarify the course of events upon ROCK depletion, we have extended the description of occurrence of each event after infection with adenovirus and depletion of ROCKs (subheading “ROCK1 and ROCK2 act redundantly to regulate actomyosin contractility and cell shape.”).

*2) In the Introduction, the authors refer to the literature by stating that previous studies on Rock have focused on cell migration. Omission of an analysis of cell migration in this study appears to be a weakness. Clearly, senescent cells will have a migration defect, but can cell migration by examined after Rock protein loss and before the onset of senescence? The manuscript would be improved by addressing this issue.*

We have included quantification of migration speed and directionality of Rock1^∆/∆^;Rock2∆/∆ cells compared to controls (Figure 2—figure supplement 1, subheading “ROCK1 and ROCK2 act redundantly to regulate actomyosin contractility and cell shape.”). This analysis was done between Days 3 and 4 after adenovirus infection, before the onset of senescence. An increase in migration speed and directionality is seen in Rock1^∆/∆^;Rock2^∆/∆^ cells compared to controls as described previously (Lomakin et al., 2015).

*3) The Trp53-independence of the senescence is not completely convincing based on the data presented. The key experiment requires 6 floxed alleles to be deleted. It is important that the ablation of Trp53 and Rock genes and loss of protein expression are documented in this study to show that Trp53 is not required for senescence.*

We have included the Western Blot analysis of cells used for SA-βGal analysis showing that p53, together with ROCK1 and ROCK2, have been depleted upon Cre recombination, showing that the infection is efficient enough to cause recombination of all 6 floxed alleles (Figure 4—figure supplement 1; subheading “ROCK regulates cell cycle proteins CKS1 and CDK1”). We also have data using P53 wild type and null MEFs that have been treated with the Rock inhibitors, H1152 and GSK269962A for 5 days followed by SA-βGal staining to show that the observed senescence phenotype upon loss of Rock is not dependent on p53 status (Figure 4—figure supplement 1; subheading “ROCK regulates cell cycle proteins CKS1 and CDK1”).

*4) The mechanism of Trp53-independent senescence is not well established* – *it may require Cdk1, CycA, and CKS1, but the mechanism is unclear and the stated hypothesis is not tested. As an initial description, this is acceptable, but these caveats should be stated. It is also unclear why the text invokes Cdk1, CycA, and CKS1 while the Abstract only refers to Cdk1 and CKS1. Moreover, in subheading “Loss of ROCK-dependent actomyosin contractility leads to cellular senescence”, the authors argue that there is a block in G1, while in the Abstract, the authors indicate that it was a multi-stage cell cycle arrest. The wording is confusing and should be clarified in the Abstract and the text of the Results section of the manuscript.*

We have re-worded the manuscript Abstract and Discussion (paragraph four) to clarify that although CKS1, Cyclin A and CDK1 are involved in the development of senescence, more work is required to determine the mechanism of how CKS1, Cyclin A and CDK1 are down-regulated to cause this phenotype. The term multi-stage cell cycle has been changed to cell cycle arrest (Abstract).

*5) The purpose of the SRM analysis of the tumor cell lines is unclear since the presence of Rock can be detected by western. What is the evidence that the tumor cells are not a mixture of Rock positive and negative cells? It appears that the presence of Rock is interpreted as if all the cells must express Rock* – *importantly, this interpretation is required for the authors' conclusions. A more rigorous test would be immunostaining of tumor sections with antibodies to Rock and/or phospho Ser-19/Thr-20 MLC if that is technically feasible.*

a) The purpose of the SRM analysis was to determine the absolute amount of the two ROCK isoforms to analyze whether one isoform was more expressed than the other which is not possible by Western Blot analysis (paragraph three, subheading “ROCK function is essential for tumorigenesis, but ROCK1 and 2 proteins act redundantly.”). Because we had observed subtle differences when knocking out individual isoforms in the two mouse tumor models, one could have suspected a higher expression of one isoform over the other. By using SRM, we have shown in vivo removal of either ROCK isoform, as well as comparing absolute quantities of ROCK1 vs ROCK2. We have added sentences within the Results section (paragraph three, subheading “ROCK function is essential for tumorigenesis, but ROCK1 and 2 proteins act redundantly”) and Discussion (paragraph six) to clarify the use and potential of SRM.

b) We spent a great deal of time assessing Rock loss in tumor and tissue sections by immunohistochemistry and immunofluorescence with antibodies to ROCK but found that these antibodies were not specific to either ROCK hence is was not been possible to monitor each isoform in situ. We also used pMYPT as a surrogate for pMLC but again found that in our hands this was not sensitive enough to quantify loss of ROCK in single cells. The cells derived from these tumors do show that there is incomplete excision of one or other of the ROCK isoforms indicating that the tumors are not a mixture of ROCK positive and negative cells but potentially a mixture of ROCK negative cells with different isoforms excised. The cells in which complete excision has taken place do not proliferate and hence do not form a tumor.

*6) Paragraph two, subheading “Depletion of both ROCK1 and 2 leads to multi-stage cell cycle arrest through defects in actomyosin contractility”. To test whether the effect of Rock depletion is caused by actomyosin contractility, the authors examined the effect of blebbistatin. Similar comments are made throughout the manuscript. This experiment is not a formal test of whether the Rock phenotype is caused by a change in actomyosin contractility – it is only a correlation that the phenotypes caused by blebbistatin and Rock depletion are similar. These statements should be re-worded throughout, including the title of the manuscript.*

We have adjusted the title, text and Discussion to highlight that even though a similar phenotype is seen comparing inhibition/depletion of ROCK, leading to a loss of actomyosin contractility, and inhibition of actomyosin contractility through myosin II using blebbistatin, we cannot be sure that these are within the same pathway. However, correlative evidence suggests that this may be the case. Changes in Abstract, Results section (paragraph one, subheading “ROCK regulates cell cycle proteins CKS1 and CDK1”) and Discussion (paragraph four).

*7) Figure 1 does not necessarily show a 'significant slower' growth rate as stated – but rather sluggish growth, perhaps caused by a delay in growth initiation. If these mice had been allowed progress further, the rate of growth might have been restored and the slope indistinguishable. In their mice experiments, was there any evidence for senescence* in vivo *at the phase of the growth curve where there was a visible difference in growth – perhaps around day 15 in Figure 1?*

a) At the time of injection, not all cells have complete excision of the Rocks (approximately 40% of cells have incomplete excision of Rock, which is based on the growth rate and retainers found to be growing out in culture). We know that a single allele of Rock is sufficient for proliferation but complete loss of ROCK will lead to senescence and an inability to proliferate, thereby leading to the ‘sluggish growth’ seen in Figure 1. We would also anticipate that the growth would be restored to wild type levels but at a longer time point because these cells (with incomplete excision) behave like wild type cells and will eventually take over the tumor.

b) We have carried out the experiment suggested above, regarding senescence in vivo. We terminated the experiment at day 15 and stained the tumors for SA-Β Gal but found that there were very few senescent cells in both the control (Rock1f/f;Rock2f/f (p53 DD; H-RasV12) tumors and tumors arising from Rock1^∆/∆^;Rock2^∆/∆^ (p53 DD; H-RasV12) MEFs and found no difference between them.

We have however carried out SA-βGal analysis in tumors generated by injection of 690cl2 mouse melanoma cells, followed by treatment of the mice with the ROCK inhibitor AT13148. This treatment resulted in a significant decrease in tumor growth and increase of SA-βGal positive cells. The data can be found in Figure 3—figure supplement 1 and within the Results section (paragraph one, subheading “Loss of ROCK-dependent actomyosin contractility leads to cellular senescence”).

8) The key blots should be quantified. Due to the different exposures and background, it is a little difficult to assess the degree of protein changes at times.

The key blots have been quantified and the quantifications can be found in Figure 2—figure supplement 1 and Figure 4.